# Adaptive Group Robust Ensemble Knowledge Distillation

**Patrik Joslin Kenfack**  *patrik-joslin.kenfack.1@ens.etsmtl.ca*
*ÉTS Montréal, Mila - Quebec AI Institute*

**Ulrich Aïvodji**  *Ulrich.Aivodji@etsmtl.ca*
*ÉTS Montréal, Mila - Quebec AI Institute*

**Samira Ebrahimi Kahou**  *Samira.Ebrahimi.Kahou@gmail.com*
*University of Calgary, Mila - Quebec AI Institute*
*Canada CIFAR AI Chair*

**Reviewed on OpenReview:** *https://openreview.net/forum?id=G2BEBaKd8Y*

## Abstract

Neural networks can learn spurious correlations in the data, often leading to performance degradation for underrepresented subgroups. Studies have demonstrated that the disparity is amplified when knowledge is distilled from a complex teacher model to a relatively "simple" student model. Prior work has shown that ensemble deep learning methods can improve the performance of the worst-case subgroups; however, it is unclear if this advantage carries over when distilling knowledge from an ensemble of teachers, especially when the teacher models are debiased. This study demonstrates that traditional ensemble knowledge distillation can significantly drop the performance of the worst-case subgroups in the distilled student model even when the teacher models are debiased. To overcome this, we propose Adaptive Group Robust Ensemble Knowledge Distillation (AGRE-KD), a simple ensembling strategy to ensure that the student model receives knowledge beneficial for unknown underrepresented subgroups. Leveraging an additional biased model, our method selectively chooses teachers whose knowledge would better improve the worst-performing subgroups by upweighting the teachers with gradient directions deviating from the biased model. Our experiments on several datasets demonstrate the superiority of the proposed ensemble distillation technique and show that it can even outperform classic model ensembles based on majority voting. Our source code is available at https://github.com/patrikken/AGRE-KD

## 1 Introduction

When trained with empirical risk minimization (ERM), neural networks are susceptible to capturing spurious correlations in the data (Tiwari & Shenoy, 2023), which are features that correlate with but not are causally related to the class label (Qiu et al., 2023). In particular, the class label might spuriously correlate with patterns in the data that are easier to learn than the intended pattern. For example, in the Waterbirds dataset (Sagawa et al., 2019), which contains images of landbirds and waterbirds, most landbirds are located in a land background, and waterbirds in a water background. Instead of predicting the actual bird species, the model trained with ERM can achieve high accuracy by relying on the background to make predictions. This results in a significantly higher error for underrepresented subgroups that do not exhibit the spurious correlation (e.g., landbird on water background and waterbird on land background). Several works have shown that model compression methods such as pruning (Hooker et al., 2020), and knowledge distillation (Lukasik et al., 2023; Lee & Lee, 2023; Wang et al., 2023) can exacerbate the performance disparities between different subgroups. In knowledge distillation (KD), a network with a smaller capacity (student) is trained using the output of a higher capacity network (teacher) (Hinton et al., 2015). While KD can improve the student model's average performance, the gain is not uniform across subgroups (Lukasik et al., 2023).

On the other hand, deep ensemble models have been shown to enhance generalization performance compared to individual models (Ganaie et al., 2022), and simple deep ensemble models with the same architecture, objective, and optimization settings can attenuate this shortcoming and improve the worst-case group performance (Ko et al., 2023; Kenfack et al., 2021). However, evaluating several models at test time can be computationally expensive, making them less practical for deployment on edge devices. To address this issue, ensemble knowledge distillation involves distilling the knowledge of multiple teachers to a single student model (You et al., 2017; Radwan et al., 2024), and it remains unclear whether distilling from an ensemble of teachers improves the student's worst-group performance.

This paper studies how knowledge distillation from multiple teachers impacts underrepresented subgroups. We investigate whether the subgroup performance gains of deep ensemble models apply to ensemble knowledge distillation. Focusing on logit distillation, we consider ERM-trained (biased) and debiased teachers obtained by deep feature reweighting (DFR). DFR is a method proposed by Kirichenko et al. (2022), showing that simply retraining the last layer of the ERM-trained model using a small held-out validation set of group-balanced data can mitigate the spurious correlation. In this work, we also investigate whether the student model can learn debiased representations from the teacher's output, whose only the last layer was retrained to mitigate bias.

The rationale behind focusing on logit distillation for investigating bias in ensemble knowledge is also motivated by the study by Izmailov et al. (2022), where the authors demonstrated that several debiasing methods do not learn better feature representation but correctly weight core features in the last classification layer, which reduce the reliance on spurious features for predictions. Our results reveal that underrepresented subgroups can be negatively impacted when training the student model using the aggregated outputs of multiple teachers. Other ensemble distillation approaches have been designed to boost the student's performance by modelling a better aggregation of the teachers' knowledge (Du et al., 2020; Zhang et al., 2022). In particular, the ensemble distillation method proposed by Du et al. (2020) aims to find a better compromise when teachers have conflicting predictions, and the method by Zhang et al. (2022) ensures distillation is done only using teachers with confident predictions. Our results show that these ensemble distillation methods do not effectively fix the performance disparity across groups in the student model.

We therefore propose an Adaptive Group Robust Ensemble Knowledge Distillation (AGRE-KD) method to encourage the student to improve the performance of unknown worst-case subgroups. Specifically, our method relies on a model that has captured the spurious correlation (i.e., a biased model) to guide the teachers' outputs aggregation process and ensure the student model does not capture biased knowledge from teachers. Prior work has relied on a reference classifier to target and upweight samples from unknown worst-case, using the errors of the reference classifier (Liu et al., 2021; Nam et al., 2020) or its per-sample gradient magnitude (Ahn et al., 2022). In contrast, our proposal uses the gradient direction of the biased model to select and weigh teachers' outputs adaptively during the training.

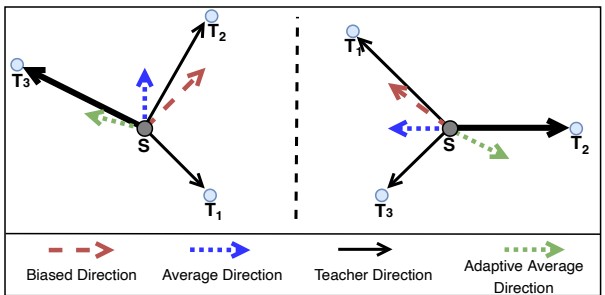

Figure 1: Illustration of our adaptive weighting method based on gradient direction. The bolder lines indicate the teacher's higher weight in the aggregated output.

Intuitively, training the student model solely following the gradient direction that minimizes its KD loss with a biased teacher can result in local/global minima with a higher loss for the underrepresented groups (Lukasik et al., 2021). Our results suggest this behaviour can be exacerbated in ensemble knowledge distillation since the inherent consensus from the teachers' gradient direction can be dominated by a direction that minimizes the average error at the expense of the worst-group error. Our proposed methods compare the gradient direction of student loss with the biased model and each teacher in the ensemble, and upweight teachers whose gradient direction deviates the most from the biased model. Figure 1 illustrates the gradient directions of the student loss with the biased model and three teachers and shows how classic averaging yields a direction that

mostly aligns with the biased teachers, while our adaptive weight prioritizes the least biased models. As the gradient magnitude can be very noisy, we compare gradient directions by computing the dot product of their normalized vectors (i.e., the cosine similarity), which removes the influence of the gradient magnitude. The contributions of this paper can be summarized as follows:

- We demonstrate empirically that ensemble knowledge distillation amplifies the performance drop for the worst-case group, contrary to deep ensemble models. We attribute this to the reduced capacity of the student network by showing that ensemble self-distillation using models with the same capacity improves robustness to spurious correlation and achieves comparable performance with the teachers.

- We show that distilling knowledge from teacher models with only their last layer retrained to mitigate bias provides significantly more robust knowledge to the student model.

- We propose a novel gradient-based weighting scheme to ensure the distillation process minimizes the student's loss towards better worst-case group error. The proposed method utilizes a model that has learned the spurious correlation (biased model) to orchestrate the distillation process in the gradient space.

- We perform intensive experiments on four well-known benchmarks, and the results demonstrate the superiority of the proposed method.

## 2 Background

We consider a multiclass classification task using a given the training $D = \{(x_i, y_i, a_i)\}_{i=1}^m$, where $x_i \in \mathcal{X}$ is the input feature, $y_i \in \mathcal{Y}$ the target variable with $c = |\mathcal{Y}|$ classes, and $a_i \in \mathcal{A}$ an unknown spurious feature. We aim to build a classifier $h(\cdot)$ that accurately predicts the class of the unlabelled test dataset. The classifier uses mapping function $f : \mathcal{X} \to \mathbb{R}^c$, that assigns scores $[\sigma_1(z_1), \ldots, \sigma_c(z_c)]$, such that $z$ is the output logits of the given sample $x$ and $\sigma_y(z)$ the softmax function defined as $\sigma_y(z) = \frac{\exp(z_y)}{\sum_{j \in [c]} \exp(z_j)}$, $\forall y \in [c]$.

The classifier is derived by predicting the class label that maximizes the softmax, $h(x) = \mathrm{argmax}_{j \in [c]} \sigma_j(z)$. We evaluate the classifier's performance during training using a loss function, such as the softmax cross-entropy loss function, which measures how accurately samples are classified. When the attribute $a$ spuriously correlates with the target $y$, ERM-trained models can strongly rely on the spurious feature and fail to generalize on the subgroups where the spurious correlation does not hold (Ye et al., 2024).

**Knowledge Distillation (KD).** In KD, a student network $f^s$ aims to achieve performance close to the higher-capacity network by mimicking the teacher model $f^t$ (Hinton et al., 2015). In practice, we train the student model to mimic the teacher's output by minimizing the Kullback-Leibler (KL) divergence between their outputs, defined as follows:

$$\mathcal{L}_{\mathrm{KD}} = \tau^2 \cdot \mathrm{KL}(\sigma(\frac{z^s}{\tau}) \, , \, \sigma(\frac{z^t}{\tau})) \tag{1}$$

where $z^s$ and $z^t$ are the student and the teacher logits, respectively. $\tau$ is the temperature parameter controlling the smoothness of the probability distribution for more fine-grained information. The student loss can be combined with the classification loss on the ground truth label using the following double loss,

$$\mathcal{L} = \alpha \cdot \mathcal{L}_{\mathrm{KD}} + (1 - \alpha) \cdot \mathcal{L}_{\mathrm{cls}} \tag{2}$$

where $\mathcal{L}_{\mathrm{cls}}$ is the classification loss (e.g., cross-entropy loss) between the student's output and the ground truth label ($y$), and $\alpha$ is a hyperparameter controlling the classification loss and knowledge distillation loss. In other KD techniques, the student network is enforced to mimic the teacher's internal representation instead of only outputs (Romero et al., 2014). Transferring knowledge from intermediary representation (feature-level) can provide more fine-grained information and boost the students' performance (Romero et al., 2014). On the other hand, recent studies have shown that models trained with ERM still learn core features while spurious features are only amplified in the last classifier layer (Kirichenko et al., 2022; Qiu et al., 2023; Izmailov et al., 2022). In this regard, we restrict ourselves to logit distillation and leave feature distillation for future exploration.

**Deep Ensemble.** Ensemble learning is a widely used technique to boost the generalization performance of a model (Allen-Zhu & Li, 2020). Studies have shown that training several independent models and averaging their predictions at test time can yield a model that outperforms each individual model in the ensemble (Ganaie et al., 2022). However, evaluating multiple models for predictions at test time limits the practical use of deep ensembles due to computational overhead. Knowledge distillation can address this issue by enforcing the student network to mimic the ensemble's output.

**Ensemble KD.** Like in ensemble learning, distilling knowledge from multiple teachers instead of a single one is expected to improve the student model (You et al., 2017). In ensemble KD, we train the student model using the averaged softened output or the averaged knowledge distillation losses of $M$ teachers[1] as follows:

$$\mathcal{L}_{\text{ensKD}} = \tau^2 \cdot \text{KL}(\sigma(\frac{z^s}{\tau}) \, , \, \frac{1}{M} \sum_{m=1}^{M} \sigma(\frac{z^m}{\tau})) \tag{3}$$

This ensemble KD loss can be plugged in equation 2 for training the student with KD from an ensemble of $M$ teachers. In contrast to these methods, our method is fully unsupervised, both based in terms of ground truth class label and group information. Related works more aligned with our approach are unsupervised methods that leverage teachers' diverse knowledge to improve students' performance. (You et al., 2017; Du et al., 2020; Fukuda et al., 2017; Zhang et al., 2022; Kwon et al., 2020). For example, Fukuda et al. (2017) suggest that randomly selecting a teacher during mini-batch training can allow the student model to capture complementary knowledge of teachers. Other authors argue that simply averaging teachers' softmax outputs can mislead the student model, particularly when there is competition or contradictions between teachers. In this regard, Zhang et al. (2022) proposes a sample-wise weighting for teacher loss based on the confidence of the teacher's prediction compared to the ground label. Other methods consider label-free weighing schemes by comparing teachers in the gradient space. For instance, Du et al. (2020) use a multi-objective optimization approach in the gradient space to find teachers that agree the most in the gradient direction that minimizes their loss with the student model. Similarly, Zhou et al. (2024) sample-wise teachers selection by only averaging the majority of teachers with the same gradient directions. Our method also operates within gradient space to enhance students' resilience to spurious correlation. Our experiments demonstrate that adhering to the majority of teachers' opinions does not always benefit the underrepresented subgroups.

**Spurious Correlation** Let $\mathcal{D}_{tr} = \{(x_i, y_i)\}_{i=1}^n$ be the training dataset, where each input $x_i \in \mathcal{X}$ belongs to one of $K$ classes $y_i \in \mathcal{Y}$. Along with the label $y_i$, each sample also contains a spurious attribute $a_i \in \mathcal{A}$, which is correlated with but not causally predictive of $y_i$. A spurious correlation, denoted $\langle y, a \rangle$, refers to the association between a class $y \in \mathcal{Y}$ and a spurious attribute $a \in \mathcal{A}$. In practice, an attribute $a$ may be linked to multiple classes, i.e., $\phi : \mathcal{A} \mapsto \mathcal{Y}^{K'}$ for some $1 < K' \leq K$, depending on the training data $\mathcal{D}_{tr}$. To capture this structure, each sample can be assigned a group label $g = (y, a)$, and the set of all such group labels is $\mathcal{G} = \mathcal{Y} \times \mathcal{A}$. For example, within $\mathcal{D}_{tr}$, the same attribute $a$ might appear in two different classes, giving rise to correlations $\langle y, a \rangle$ and $\langle y', a \rangle$. A model trained naively on this data may incorrectly rely on $a$ to predict $y$, which will lead to systematic errors when encountering samples with $\langle y', a \rangle$ (Ye et al., 2024). As the average accuracy does not fully capture the robustness of the model to spurious correlation, we use the worst-group accuracy (WGA) to measure the model's reliance on spurious correlation for predictions. Furthermore, in the presence of spurious features, a model $f$ parametrized by $\theta$ is optimized to minimize the loss of the worst-performing subgroup, i.e.,

$$\arg \min_{\theta} \max_{g \in \mathcal{G}} \frac{1}{|g|} \sum_{i \in g}^{|g|} \mathcal{L}(f(x_i; \theta), y_i) \tag{4}$$

The WGA therefore quantifies the accuracy of the group experiencing the smallest test accuracy. Another line of related work aims to make the model robust to distribution shift (Yang et al., 2024), especially covariate shift. Unlike spurious correlation, covariate shift occurs when the training and test data follow different

---

[1]Du et al. (2020) showed that the averaged softened output is equal to averaged KL loss.

distributions, while the conditional distribution of output values given input points is unchanged. In other words, the marginal distribution of features changes, but the conditional relationship between labels and features remains unchanged. Techniques like importance weighting or domain adaptation aim to correct for this shift (Sugiyama et al., 2007), while we focus in this work on addressing model reliance on spurious features during training. Both notions could be related if the 'spurious feature' is the source of the covariate shifts, i.e., a feature that was predictive for the training data becomes spurious in the test set; however, such scenarios are less likely to occur since the model should not rely on spurious features.

## 3    Related Work

**Bias mitigation without group label.**    We consider settings where samples in the dataset $D$ are associated with *unknown* group labels that spuriously correlate with the class label. For example, in the CelebA dataset, the class 'hair color' (blond, non-blond) correlates with the gender (male, female) since most images with blond hair belong to the female group. Neural networks can capture this correlation, resulting in poor performance for certain subgroups (e.g., blond males) (Sagawa et al., 2019). We aim to ensure that the model does not capture spurious correlations in the data and accurately classifies samples from all subgroups. In particular, we measure the model's bias using the *worst-case group accuracy* (WGA), i.e., group of samples where the spurious correlation does not apply. Several methods have been proposed to mitigate these biases in individual models (Kenfack et al., 2024a; Ye et al., 2024). When the group information is known, Sagawa et al. (2019) propose Group Distributionally Robust Optimization (DRO) that minimizes the loss of the group experiencing the maximum loss. However, group information can be costly to collect or unavailable due to privacy restrictions (Kenfack et al., 2024b). Several methods have been proposed in this setting to improve the worst-case group performance without group labels (Ye et al., 2024). For instance, Liu et al. (2021) and Nam et al. (2020) rely on a reference classifier to target and upweight worst-performing subgroups. These methods use the mistakes of the reference classifier to improve group robustness by up-weighting misclassified samples (Liu et al., 2021). The reference classifier is generally trained to amplify the misclassification of samples from the unknown worst-performing group (Nam et al., 2020). While these methods do not use group labels during the training, they require access to a small validation set with group labels for model selection and intensive hyperparameter tuning (Kenfack et al., 2024a). Kirichenko et al. (2022) proposed *Deep Feature Reweighting* (DRF) for training group robust model by retraining the last layer of ERM model with a small subset of a held-out group-balanced dataset, which achieves state-of-the-art worst-case group performance. However, all debiasing methods use neural networks of high capacity, and they are less effective on models with smaller capacity Izmailov et al. (2022). Furthermore, the effect of distilling these debiased models on the student model's robustness remains underexplored. This work builds on Kirichenko et al. (2022) for debiasing the teacher models by retraining their last layers using a held-out group balanced with fewer group annotations, and we investigate whether distilling knowledge from an ensemble of (debiased) teachers can lead to more robust student models.

**Bias in Knowledge Distillation.**    Several works have studied bias in knowledge distillation with a single teacher model (Lukasik et al., 2023; Lee & Lee, 2023; Lukasik et al., 2023; Tiwari et al., 2024; Bassi et al., 2024). In particular, Lukasik et al. (2023) demonstrates that teacher errors can be amplified by the student during distillation and proposed a mitigation strategy that distills only the confident predictions of the teacher. However, their study focuses on worst-class errors and KD with a single teacher, while we study worst-subgroup errors with multiple teachers. Lee & Lee (2023) propose an adapted version of *Simple knowledge distillation* (SimKD (Chen et al., 2022)) that transplants the last layer of the teacher to student and only distills features. With the teacher trained with Group DRO, they show that transplanting the teacher's last layer to the student only improves the worst-case group performance if the feature distillation is performed by upweighting misclassified sample from a reference classifier (Lee & Lee, 2023). However, their method uses the reference classifier by Liu et al. (2021), where the efficiency of sample weighting requires intensive tuning of the number of epochs to train the reference classifier. Similarly, Tiwari et al. (2024) uses the earlier layers of the neural network to train a reference classifier and shows that it improves the recall of worst group samples within the misclassification set, which are upweighted in the KD loss. However, their method also requires class labels to derive the misclassified samples. In Lukasik et al. (2023), the authors study where it is best to apply the debiasing mechanism (Group DRO) and conclude that applying the robust

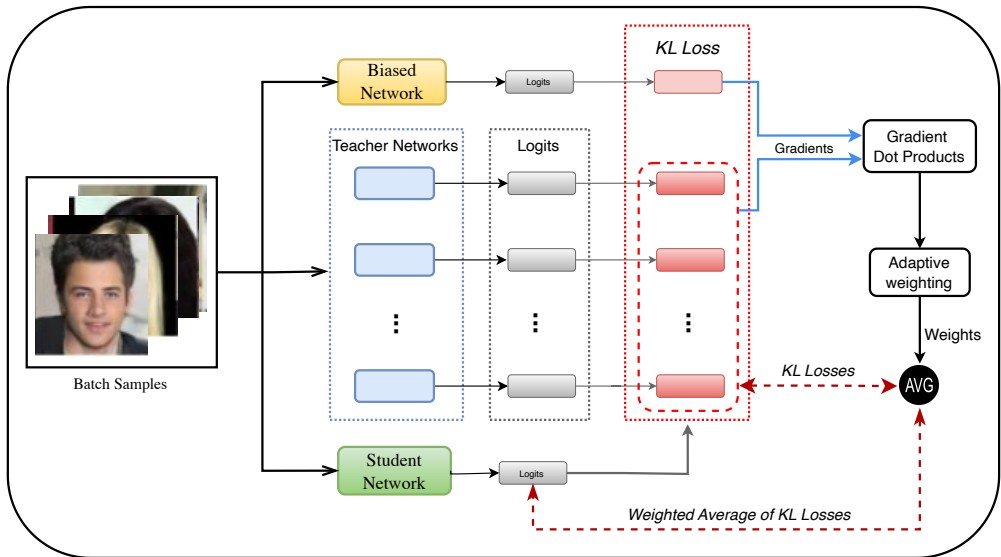

Figure 2: Overview of AGRE-KD.

loss to both the teacher and student model improves the average performance along with the worst-case group performance. In contrast to these prior works, we study bias in knowledge distillation with multiple teachers **without group information and class labels**. We investigate whether the subgroup's performance gain observed in deep ensemble models also applies when the knowledge of the ensemble is distilled to a single model. We aim to achieve better worst-case performance across subgroups when aggregating the outputs of multiple teachers in knowledge distillation. To the best of our knowledge, this represents the first study on bias in ensemble knowledge distillation.

## 4 Adaptive Group Robust Ensemble Knowledge Distillation (AGRE-KD)

In this work, we considered each teacher in the ensemble to have the same architecture and trained using different random initializations. Prior work has shown ensemble models with different random initializations are diverse enough to improve the performance (Ganaie et al., 2022). The first contribution of this work is showing that, unlike deep ensemble models that can improve the WGA (Ko et al., 2023), ensemble knowledge distillation tends to amplify bias in the distilled model.

To address this problem, we propose AGRE-KD, an adaptive ensembling knowledge distillation strategy that ensures the student model captures robust knowledge from the teachers. AGRE-KD relies on a model pretrained with ERM that captured the dataset's spurious correlation (biased model). Intuitively, suppose a student model takes gradient steps toward the direction that minimizes its KD loss with the *biased* model. In that case, the resulting student model will likely capture and even amplify the reliance on the spurious correlation, i.e., the local/global minimum in that direction likely provides the worst performance for the underrepresented subgroups. Following this intuition, we derive a weighting scheme that prioritizes teacher models whose gradient direction disagrees with the biased ERM model. Figure 2 provides an overview of our proposed method.

**Teacher training.** We train each teacher model in the ensemble independently using standard ERM and cross-entropy loss with the ground truth class labels. Teacher models have the same architecture and hyperparameters and only differ in random seeds used for weights initialization; prior studies have shown that independent training with different random weight initializations can provide sufficient ensemble diversity to improve the performance of the model ensemble (Allen-Zhu & Li, 2020; Ganaie et al., 2022). As aforementioned, we obtain debiased teachers using the approach introduced by Kirichenko et al. (2022) to retrain the last layer of the ERM model on a small proportion of the held-out group-balanced dataset. We

perform the retraining step of the last layer using group-balanced batch sampling instead of the averaging models trained over group-balanced subsets of the data (LaBonte et al., 2024). This debiasing process is very simple and computationally inexpensive since it involves training a logistic regression model on a smaller dataset. Following Kirichenko et al. (2022); LaBonte et al. (2024), we use half of the validation set of each benchmark to perform last-layer retraining with DFR; as we do not use the group and class labels during the KD training, we do not perform further hyperparameter tuning or model selection. After training, teacher models are frozen, and their output is aggregated to train student models as illustrated in figure 2.

**Biased model.** For the *biased* model used by our method to compute teachers' weights, we randomly select one teacher model trained with ERM. ERM-trained models are known to be sensitive to spurious correlation (Kirichenko et al., 2022; Ye et al., 2024). This provides us with intuition that the gradient direction updates of a model during ERM training are biased towards the majority, and leverage this intuition to upweight teacher models that deviate from the direction provided by an ERM model. Moreover, as we show via an ablation study in Supplementary D (Table 5), the performance of our gradient-based weighting scheme remains robust to different choices of ERM model architectures.

**Gradient-based weighting scheme.** For a given minibatch, we compute the student KD loss regarding the biased model $b$ and each teacher $t$. Before aggregating the teachers' outputs, we compute sample weights based on the similarity between the gradient direction of each teacher and the biased model. Consider $\ell_i^t(\theta)$ the KD loss on sample $i$ regarding the $t$-teacher, and $\ell_i^b(\theta)$ KD loss regarding the biased model, with $\theta$ the parameters of the student model. The dot product ($\langle \cdot, \cdot \rangle$) between the *normalized*[2] gradients of the student KD loss with the teacher and the biased model indicates which teachers align with the biased model in gradient space. In particular, when $\langle \nabla \ell_i^t(\theta), \nabla \ell_i^b(\theta) \rangle > 0$ both models have the same gradient directions and their gradients are in the opposite direction when $\langle \nabla \ell_i^t(\theta), \nabla \ell_i^b(\theta) \rangle < 0$. In extreme cases, when the dot product of the gradients is closer to 1, both models have precisely the same directions; if the dot product is close to $-1$, the models are in opposite directions. Thus, to penalize teachers with a gradient direction closer to the biased model, we downweight teachers' output for samples having dot products get closer to 1 and upweight samples as their dot products get closer to $-1$. More formally, we compute the sample-wise teacher's weight as follows:

$$W_t(x_i) = 1 - \langle \nabla \ell_i^t(\theta), \nabla \ell_i^b(\theta) \rangle \tag{5}$$

**Adaptive knowledge distillation.** The weighting scheme in Equation 5 suggests that teacher models who behave similarly to the biased model in the gradient space will have less influence on the aggregated outputs. Therefore, we aggregate the teacher's outputs using the sample-wise weighted average KD (wKD) loss defined as follows:

$$\mathcal{L}_{\text{wKD}} = \frac{W_t(x_i) \cdot \mathcal{L}_{\text{KD}}}{\sum_t W_t} \tag{6}$$

And we train the student model with the following final loss

$$\mathcal{L} = \alpha \mathcal{L}_{\text{wKD}} + (1 - \alpha)\mathcal{L}_{\text{cls}} \tag{7}$$

As aforementioned, we focus on unsupervised knowledge distillation, i.e., we set $\alpha = 1$ to investigate the robustness of the "dark knowledge" provided by an ensemble of eventually debiased teachers. Supplementary A provides a detailed algorithm of the training process of AGRE-KD. Note that the use of an additional biased model to orchestrate the distillation process does not add extra complexity compared to related work using a reference classifier to identify the worst-performing subgroups (Nam et al., 2020; Liu et al., 2021; Kenfack et al., 2024a). Furthermore, as we will see in the experiments, our proposed method is robust to the choice of biased model architecture trained using ERM without.

---

[2]Table 6 in the Supplementary shows the importance of ignoring the magnitude of the gradients.

## 5   Experimental Results

In this section, we present the experimental setup and the empirical results. We experiment with several ensemble KD techniques and analyze their impact on the worst-group accuracy of the student model. We compare our adaptive ensemble knowledge distillation to these methods and demonstrate its effectiveness in improving the student model's worst-group performance.

### 5.1   Setup

We evaluate the worst-case performance of the proposed method on four classification tasks: one synthetic dataset (Colored MNIST), two real-world datasets from the vision domain (Waterbirds and CelebA datasets), and one real-world dataset from the language domain (CivilComments).

- **Colored MNIST** following Nam et al. (2020); Li & Vasconcelos (2019), we alter the MNIST dataset (LeCun et al., 2002), designed for digit classification, by artificially adding a Color attribute. To define the Color, we randomly select ten distinct RGB values and fix them across all experiments. Each of these values serves as the mean of a 3-dimensional Gaussian distribution, forming ten distinct Color distributions. To create correlations between digits and colors, we pair each digit with one of these distributions. A bias-aligned sample is generated by coloring a digit with an RGB value drawn from its paired distribution, while a bias-conflicting sample is produced by sampling from any of the remaining nine color distributions (Nam et al., 2020). Under this setup, a model that captures the spurious correlation will predict color instead of the digits.

- **Waterbirds** (Sagawa et al., 2019; Liu et al., 2021) is a dataset of birds derived from Caltech-UCSD Birds (CUB) (Wah et al., 2011) by synthetically creating a spurious correlation between bird species and the background. In particular, the class label is the type of bird appearing in the image (waterbirds, landbirds), and the background landscape (water, land) spuriously correlates with the bird type. Here, the minority subgroups represent images with the background landscape not aligned with the bird type, i.e., {`waterbird`, `land background`} and {`landbird`, `water background`}.

- **CelebA** (Liu et al., 2015) dataset contains images of celebrities with 40 facial attributes. In this dataset, the attribute `hair color` is spuriously correlated with `gender`. We consider hair color {`blond`, `non-blond`} as the class label and gender {`male`, `female`} as group information.

- **CivilComments** (Koh et al., 2021) is a textual dataset collected from online comments. The task is to predict whether a comment is `toxic` or `non-toxic`. The label is spuriously correlated with comments related to some demographic subgroups such as gender (male, female), race (white, black), and sexual orientation (LGBT). We consider a binary indicator of comments related to these demographic subgroups as spurious group information.

**Network Architecture and Training.**   For the Colored MNIST dataset, we use MobileNetV2 (Sandler et al., 2018) for the teacher models and a simple CNN with two convolutional layers as for the student model. For the real-world vision tasks (CelebA and WaterBirds), following prior work (Tiwari et al., 2024; Lee & Lee, 2023), we use the Resnet-18 (He et al., 2016) architecture for the student model and the Resnet-50 (He et al., 2016) architecture for the teacher models. Both networks are pretrained on the ImageNet-1K (Russakovsky et al., 2015) dataset. For the language task, we use the BERT (Devlin et al., 2019) model for teachers and the DistilBERT (Sanh et al., 2019) for the student model; and language models are pretrained on Book Corpus and English Wikipedia. Following related works on KD (Du et al., 2020; Fukuda et al., 2017; Chen et al., 2022), we set the temperature hyperparameter $\tau = 4$ (Eq. 1) and show in an ablation study in Supplementary D (Figure 5) that increasing $\tau$ can exert positive effect on WGA up to certain values. We provide further details about hyperparameters in the Supplementary B.

**Baselines.**   In addition to the standard training process using the one-hot class label for training each teacher model (Section 2), we consider other ensemble knowledge distillation methods aiming to improve the student's performance. In particular, we consider the following baseline:

- **Deep Ensemble** (Deep Ens.): This baseline corresponds to deep ensembling using a majority voting scheme of models with the same capacity as the student model. In particular, given a set of models, the predicted class label represents the class that received the most votes for models in the ensemble.

- **One-hot**: A student model trained with ground truth labels without knowledge distillation.

- **Random** (Fukuda et al., 2017): During each mini-batch training, this method randomly selects a teacher model from the ensemble to train the student model. Fukuda et al. (2017) referred to this technique as *switched-training* as the weights of the student model are updated by switching across teacher labels at the minibatch level.

- **AVER** (KD with averaged teachers' outputs (You et al., 2017)): Here, we perform standard knowledge distillation following equation 3 by minimizing the KL loss between the student's softmax output and the averaged softened outputs (dark knowledge) from teachers.

- **AE-KD** (Du et al., 2020): This method is an adaptive ensembling distillation technique closest to ours. However, the method postulates that when teachers have conflicting gradient directions, a multi-objective optimization problem is solved to select the gradient direction (teachers) that satisfies most of the teachers in the ensemble.

## 5.2  Results

We train each model using three random seeds and report the means and standard deviations. We consider ensemble distillation with ten teacher models randomly sampled from a poll of pretrained (*biased*) teachers across independent random seeds. In addition to the teacher's performance and deep ensemble approach, we report the performance of the "student" trained only using the ground truth class labels (One-hot), i.e., without knowledge distillation. We report the average accuracy and WGA. We consider models trained using ERM or with last-layer retraining for debiasing. Specifically, for ensemble knowledge distillation methods, we train the student model with *biased teachers* (ERM model) and *debiased teachers* (last layer retrained with DFR (Kirichenko et al., 2022)).

### 5.2.1  Results on the synthetic dataset

We considered different settings of the Colored MNIST (CMNIST) dataset by varying the ratios of bias-aligned samples in the training dataset, i.e., the proportion of samples where the color and digit correspond $\{99.5\%, 99\%, 98\%, \text{and } 95\%\}$. This means for configuration with a ratio of 99.5%, only 0.005% of samples will have a digit-color mismatch, and decreasing the ratio reduces the strength of the spurious correlation in the training dataset. We train every baseline under each bias ratio configuration and report the average and worst-group test accuracy.

Table 1 presents the performance of different approaches on Colored MNIST under varying levels of spurious correlation. Several important observations can be drawn.

The teacher models illustrate the critical role of debiasing. Without debiasing, teachers perform poorly on the minority group (WGA as low as 30.5% under 99.5% spurious correlation), even average accuracy appears lower (37.4%) since the test set mainly contains samples where spurious correlation does not apply. Once debiased, however, teacher models show clear improvements in WGA across all regimes, confirming that debiased supervision is key to counteracting extreme spurious correlations.

Deep Ensembles models already provide a substantial performance boost: simply averaging biased models significantly boosts robustness compared to a single teacher, leading to WGA improvements of 20–40 points in extreme regimes (e.g., 29.2% to 69.8% at 99.5%). Interestingly, when ensemble predictions are used for knowledge distillation (AVER, AEKD, or AGRE-KD with biased teachers), performance patterns diverge. In highly biased settings (CMNIST-99.5%, CMNIST-99%), ensemble KD tends to underperform relative to the raw Deep Ensemble, often worsening WGA (e.g., AVER and AEKD around 49–54% WGA vs. 69.8% for Deep Ens. in CMNIST-99.5%). However, in less extreme regimes (CMNIST-98% and CMNIST-95%), ensemble KD begins to catch up and even surpass the teacher ensemble, suggesting that distillation stabilizes and refines knowledge transfer when correlations are milder.

| Models | Debiased | KD | CMNIST-99.5% | | CMNIST-99% | | CMNIST-98% | | CMNIST-95% | |
|---|---|---|---|---|---|---|---|---|---|---|
| | | | Average | WGA | Average | WGA | Average | WGA | Average | WGA |
| Teacher | ✗ | — | $37.4_{\pm0.69}$ | $30.5_{\pm0.61}$ | $55.3_{\pm0.92}$ | $50.4_{\pm1.00}$ | $71.7_{\pm0.45}$ | $68.7_{\pm0.56}$ | $87.7_{\pm0.10}$ | $86.3_{\pm0.14}$ |
| | ✓ | — | $58.4_{\pm1.84}$ | $54.7_{\pm2.31}$ | $66.0_{\pm0.97}$ | $63.1_{\pm1.00}$ | $80.3_{\pm0.60}$ | $78.1_{\pm0.70}$ | $83.4_{\pm0.53}$ | $81.7_{\pm0.58}$ |
| Deep Ens. | ✗ | — | $36.2_{\pm0.38}$ | $29.2_{\pm0.44}$ | $56.2_{\pm0.25}$ | $51.4_{\pm0.22}$ | $74.4_{\pm0.09}$ | $71.5_{\pm0.15}$ | $89.6_{\pm0.09}$ | $88.4_{\pm0.12}$ |
| | ✓ | — | $72.8_{\pm0.27}$ | $69.8_{\pm0.23}$ | $77.7_{\pm0.11}$ | $75.3_{\pm0.09}$ | $84.2_{\pm0.16}$ | $82.5_{\pm0.19}$ | $90.0_{\pm0.09}$ | $88.9_{\pm0.09}$ |
| One-hot | ✗ | ✗ | $26.9_{\pm6.52}$ | $19.0_{\pm7.44}$ | $29.3_{\pm6.94}$ | $21.6_{\pm7.93}$ | $34.5_{\pm7.93}$ | $27.4_{\pm9.01}$ | $47.0_{\pm7.91}$ | $41.4_{\pm8.97}$ |
| | ✓ | ✗ | $53.6_{\pm6.29}$ | $52.6_{\pm5.91}$ | $51.9_{\pm5.68}$ | $51.7_{\pm5.63}$ | $81.5_{\pm1.58}$ | $80.0_{\pm1.82}$ | $61.8_{\pm4.83}$ | $60.9_{\pm4.37}$ |
| Random | ✗ | ✓ | $25.0_{\pm2.41}$ | $16.7_{\pm2.58}$ | $26.5_{\pm3.31}$ | $18.5_{\pm3.58}$ | $29.4_{\pm4.24}$ | $21.7_{\pm4.61}$ | $41.1_{\pm3.92}$ | $34.6_{\pm4.29}$ |
| | ✓ | ✓ | $53.2_{\pm5.35}$ | $48.5_{\pm6.02}$ | $49.7_{\pm5.90}$ | $44.5_{\pm6.49}$ | $39.3_{\pm4.07}$ | $32.6_{\pm4.45}$ | $41.8_{\pm4.67}$ | $35.5_{\pm5.12}$ |
| AVER | ✗ | ✓ | $33.9_{\pm0.48}$ | $26.5_{\pm0.51}$ | $51.6_{\pm0.14}$ | $46.2_{\pm0.21}$ | $72.2_{\pm0.59}$ | $69.1_{\pm0.66}$ | $89.3_{\pm0.47}$ | $88.1_{\pm0.54}$ |
| | ✓ | ✓ | $54.1_{\pm0.23}$ | $49.0_{\pm0.28}$ | $62.8_{\pm0.42}$ | $58.7_{\pm0.48}$ | $86.5_{\pm0.55}$ | $85.0_{\pm0.63}$ | $88.5_{\pm0.19}$ | $87.3_{\pm0.22}$ |
| AEKD | ✗ | ✓ | $36.2_{\pm0.76}$ | $\underline{29.2}_{\pm0.81}$ | $55.0_{\pm0.83}$ | $50.0_{\pm0.84}$ | $73.4_{\pm0.78}$ | $70.5_{\pm0.81}$ | $89.7_{\pm0.09}$ | $88.6_{\pm0.07}$ |
| | ✓ | ✓ | $58.8_{\pm0.80}$ | $54.5_{\pm1.29}$ | $69.4_{\pm1.56}$ | $66.3_{\pm1.65}$ | $86.6_{\pm0.65}$ | $85.1_{\pm0.68}$ | $88.1_{\pm0.45}$ | $86.9_{\pm0.53}$ |
| AGRE-KD | ✗ | ✓ | $\underline{36.3}_{\pm0.65}$ | $29.2_{\pm0.65}$ | $\underline{55.5}_{\pm0.56}$ | $\underline{51.5}_{\pm0.59}$ | $\underline{74.2}_{\pm0.21}$ | $\underline{71.4}_{\pm0.26}$ | $\underline{90.2}_{\pm0.31}$ | $\underline{89.1}_{\pm0.34}$ |
| | ✓ | ✓ | $\mathbf{63.9}_{\pm0.33}$ | $\mathbf{59.9}_{\pm0.37}$ | $\mathbf{70.7}_{\pm0.43}$ | $\mathbf{67.5}_{\pm0.53}$ | $\mathbf{87.5}_{\pm0.51}$ | $\mathbf{86.2}_{\pm0.57}$ | $\mathbf{89.0}_{\pm0.48}$ | $\mathbf{88.7}_{\pm0.49}$ |

Table 1: **Results on the Colored MNIST dataset**. We report the average and worst-group test accuracy (WGA) for different ratios of bias-aligned samples ($\{99.5\%, 99\%, 98\%, \text{and } 95\%\}$). The *Debiased* column indicates whether the model involves debiasing with DFR or whether teacher models are debiased when the *KD* column is checked (✓). Bolded represents the best-performing student with the debiased teacher ensemble and underlined represents the best-performing with the biased teacher ensemble.

Both AVER and AEKD, which attempt to leverage ensemble outputs in different ways, offer moderate improvements over naïve or random strategies. For example, AEKD achieves 66.3% WGA at 99% with a debiased teacher, substantially higher than One-hot or Random. However, their effectiveness remains inconsistent: in extreme regimes, they still fail to match the robustness of the Deep Ensemble, and in milder regimes, they are often outperformed by stronger approaches.

Across all bias regimes, AGRE-KD consistently yields the best performance when combined with debiased teachers. It achieves the highest WGA at every correlation level, from 59.9% under the most extreme bias (CMNIST-99.5%) to 88.7% in CMNIST-95%. Notably, AGRE-KD is the only KD method that not only closes the gap with the raw Deep Ensemble in highly biased regimes but surpasses it in less extreme regimes. Even with biased teachers, AGRE-KD matches or exceeds other KD baselines, demonstrating its robustness to spurious correlation during the distillation process.

### 5.2.2 Results on real-world datasets

Table 2 summarizes the main results of the paper on real-world datasets, from which we draw the following observations:

- **Ensemble knowledge distillation can amplify bias.** From Table 2, we first observe that deep ensembles improve worst-group accuracy (WGA) relative to single models (one-hot). For example, on Waterbirds, WGA increases from 54.1% (biased model) to 59.3% (biased deep ensemble) and from 86.7% (debiased model) to 90.0% (debiased deep ensemble). On CelebA, the jump is even larger: WGA improves from 32.3% (biased model) to 37.2% (biased deep ensemble) and from 88.9% (debiased teacher) to 90.3% (debiased deep ensemble). This confirms that ensembling stabilizes predictions and mitigates reliance on spurious features.

  However, when ensemble predictions are used for knowledge distillation, the picture changes. In highly biased regimes, ensemble KD often reduces WGA. For instance, on Waterbirds, Random KD drops WGA to 39.6%, nearly 20 points below the biased deep ensemble (59.3%). Similarly, AVER KD achieves only 46.4% WGA on Waterbirds and 28.8% on CelebA, representing losses of more than 10 points compared to their ensemble teachers. These results suggest that switching between teachers or naively averaging teachers' output during distillation amplifies bias, particularly harming

minority groups, even if average accuracy remains high (e.g., CelebA average accuracy stays around 95.5%). When comparing to the one-hot baseline, we observe mixed results, but generally, ensemble KD can increase average accuracy compared to the "One-hot" baseline, but they are also more prone to bias amplification. This reinforces our finding that while ensemble KD offers advantages, debiasing mechanisms are essential to make it competitive with simpler approaches like direct training or DFR.

- **DFR teachers lead to robust student models.** When we train the teacher models using ERM and then retrain the classification layer with the group-balanced set, the resulting student models achieve significantly better worst-case group accuracy compared to those trained only with ERM teacher models. We only train the student model using the aggregated teachers' outputs without any feature distillation. For example, on CelebA, Random KD training with debiased teachers improves WGA from 27.2% (biased) to 85.1% (debiased), a remarkable +57.9 point gain. Similar patterns hold for the other ensemble KD methods with AVER KD ($28.8\% \rightarrow 83.4\%$), AE-KD ($46.9\% \rightarrow 85.0\%$), and AGRE-KD ($55.0\% \rightarrow 87.9\%$).

These results demonstrate that the retrained classification layer can provide pseudo-label distribution (dark knowledge) that reduces students' reliance on spurious features. This shows that by mimicking the teacher's outputs, the classifier layer of the student model also downweights the spurious features in its last layers (Izmailov et al., 2022). As the temperature parameter ($\tau$) also alters the teacher's output by softening its predicted probability distribution, we provide an ablation study (Appendix C.2) showing that increasing $\tau$ positively impacts the worst-group accuracy. We also show in Appendix **??** experiments with different values of $\alpha$ (Eq. 3 & 7), controlling the balance between updating the student model with ground truth label and ensemble knowledge distillation loss. The results show that the student model gets less biased as more weight is given to the knowledge distillation loss. For example, on the Waterbirds dataset, the WGA increases linearly by up to +25% with increased values of $\alpha$.

On the other hand, the improved students' WGA across ensemble distillation methods still does not match the WGA of the debiased teachers or debiased deep ensemble. For example, on Waterbirds, AVER KD with debiased teachers achieves 82.9% WGA, 7% below the 90.9% WGA of the debiased teacher. We attribute this to the smaller capacity of the student model, which we discuss in the next experiment.

- **Our weighting scheme consistently improves WGA.** Finally, our proposed AGRE-KD achieves the most robust results across datasets. With debiased teachers, AGRE-KD attains the highest WGA in all three benchmarks: 87.9% on Waterbirds, 91.9% on CelebA, and 75.9% on CivilComments. This represents improvements of 2.9, 3.0, and 0.9%, respectively, over the best competing KD methods (AEKD or AVER). Importantly, on CelebA, AGRE-KD even outperforms the deep ensemble itself (91.9% vs. 90.3% WGA), showing that our weighting scheme extracts more bias-free knowledge than naive ensembling. When all teachers are biased, AGRE-KD still identifies and upweights the least biased members. For example, on Waterbirds, it achieves 55.0% WGA, outperforming AVER KD(46.4%) and Random KD (39.6%). On CelebA, it reaches 37.6%, slightly higher than the biased teacher baseline (37.5%). While the improvements in this setting are modest—since all teachers share similar spurious gradients—the method still avoids the large collapses observed with other KD strategies. In the Supplementary D (Table 7), we provide the group-wise accuracy of each baseline method on the Waterbirds and CelebA datasets, showing how each subgroup is impacted by each method.

### 5.2.3  Impact of the model capacity.

In this experiment, we study whether the student's network capacity is a source of the implication of spurious feature learning in ensemble KD. The capacity here represents the number of trainable parameters of the model. For example, our ResNet-50 teacher models have around 25.6M parameters, substantially larger than ResNet-18 student models with approximately 11M parameters. We perform the same experiments as previously, but using self-distillation, i.e., we use the same architecture for the student and teacher models (e.g., ResNet-50 on image tasks). Results in Table 3 show that ensemble KD methods with self-distillation

| Models | Debiased | KD | Waterbirds | | CelebA | | CivilComment | |
|---|---|---|---|---|---|---|---|---|
| | | | Average | WGA | Average | WGA | Average | WGA |
| Teacher | ✗ | — | $85.7_{\pm1.80}$ | $65.6_{\pm5.75}$ | $95.4_{\pm0.07}$ | $37.5_{\pm2.27}$ | $90.0_{\pm0.25}$ | $75.9_{\pm1.15}$ |
| | ✓ | — | $94.2_{\pm0.53}$ | $90.9_{\pm1.00}$ | $93.7_{\pm2.44}$ | $90.1_{\pm1.68}$ | $85.9_{\pm0.49}$ | $77.9_{\pm0.49}$ |
| Deep Ensemble | ✗ | — | $84.4_{\pm0.00}$ | $59.3_{\pm0.44}$ | $95.6_{\pm0.02}$ | $37.2_{\pm0.00}$ | $90.6_{\pm0.02}$ | $76.1_{\pm0.10}$ |
| | ✓ | — | $93.5_{\pm0.17}$ | $90.0_{\pm0.56}$ | $92.4_{\pm0.18}$ | $90.3_{\pm0.55}$ | $85.9_{\pm0.17}$ | $76.6_{\pm0.09}$ |
| One-hot | ✗ | ✗ | $\underline{83.8}_{\pm0.97}$ | $54.1_{\pm2.21}$ | $\underline{95.5}_{\pm0.07}$ | $32.3_{\pm2.66}$ | $90.1_{\pm0.21}$ | $\underline{75.5}_{\pm0.84}$ |
| | ✓ | ✗ | $91.4_{\pm1.31}$ | $86.7_{\pm1.85}$ | $92.3_{\pm0.39}$ | $88.9_{\pm2.90}$ | $85.7_{\pm0.47}$ | $\mathbf{76.5}_{\pm1.09}$ |
| Random | ✗ | ✓ | $80.0_{\pm0.37}$ | $39.6_{\pm3.63}$ | $95.2_{\pm0.03}$ | $27.2_{\pm0.00}$ | $90.8_{\pm0.21}$ | $75.2_{\pm0.95}$ |
| | ✓ | ✓ | $89.6_{\pm0.71}$ | $77.3_{\pm2.23}$ | $\mathbf{92.5}_{\pm0.29}$ | $85.1_{\pm0.84}$ | $\mathbf{91.0}_{\pm0.18}$ | $74.2_{\pm1.04}$ |
| AVER | ✗ | ✓ | $79.2_{\pm0.80}$ | $46.4_{\pm2.39}$ | $\underline{95.5}_{\pm0.05}$ | $28.8_{\pm0.78}$ | $\underline{90.9}_{\pm0.03}$ | $74.7_{\pm1.16}$ |
| | ✓ | ✓ | $90.8_{\pm2.44}$ | $82.9_{\pm1.23}$ | $92.4_{\pm0.25}$ | $83.4_{\pm0.45}$ | $90.8_{\pm0.07}$ | $75.0_{\pm0.73}$ |
| AE-KD | ✗ | ✓ | $81.7_{\pm0.80}$ | $46.9_{\pm7.57}$ | $95.3_{\pm0.06}$ | $30.5_{\pm1.88}$ | $90.8_{\pm0.54}$ | $73.1_{\pm3.05}$ |
| | ✓ | ✓ | $90.9_{\pm1.72}$ | $85.0_{\pm1.23}$ | $92.3_{\pm0.26}$ | $87.5_{\pm1.17}$ | $90.7_{\pm0.23}$ | $74.8_{\pm1.11}$ |
| AGRE-KD(Ours) | ✗ | ✓ | $82.2_{\pm1.37}$ | $\underline{55.0}_{\pm5.47}$ | $95.4_{\pm0.04}$ | $\underline{37.6}_{\pm0.78}$ | $89.3_{\pm3.65}$ | $74.7_{\pm3.00}$ |
| | ✓ | ✓ | $\mathbf{91.3}_{\pm0.49}$ | $\mathbf{87.9}_{\pm1.23}$ | $91.7_{\pm0.20}$ | $\mathbf{91.9}_{\pm0.71}$ | $90.2_{\pm0.49}$ | $75.9_{\pm1.75}$ |

Table 2: **Comparison of ensemble KD methods**. We report the average and worst-group test accuracy (WGA) on each dataset. The *Debiased* column indicates whether the model involves debiasing with DFR or whether teacher models are debiased when the *KD* column is checked (✓). Bolded represents the best-performing student with the debiased teacher ensemble and underlined represents the best-performing with the biased teacher ensemble.

| Models | Debiased | KD | Waterbirds | | CelebA | |
|---|---|---|---|---|---|---|
| | | | Average | WGA | Average | WGA |
| Deep Ensemble | ✗ | — | $84.4_{\pm0.00}$ | $59.3_{\pm0.44}$ | $95.6_{\pm0.02}$ | $37.7_{\pm0.00}$ |
| | ✓ | — | $93.5_{\pm0.17}$ | $90.0_{\pm0.56}$ | $92.4_{\pm0.18}$ | $88.8_{\pm0.55}$ |
| One-hot | ✗ | ✗ | $\underline{85.7}_{\pm1.80}$ | $65.6_{\pm5.75}$ | $95.4_{\pm0.07}$ | $37.5_{\pm2.27}$ |
| | ✓ | ✗ | $92.2_{\pm0.53}$ | $90.9_{\pm1.00}$ | $92.5_{\pm0.34}$ | $88.2_{\pm1.68}$ |
| Random | ✗ | ✓ | $83.5_{\pm0.68}$ | $64.0_{\pm2.88}$ | $95.5_{\pm0.00}$ | $35.8_{\pm1.17}$ |
| | ✓ | ✓ | $92.8_{\pm0.49}$ | $90.3_{\pm0.23}$ | $\mathbf{92.6}_{\pm0.26}$ | $87.9_{\pm0.64}$ |
| AVER | ✗ | ✓ | $83.5_{\pm0.75}$ | $63.7_{\pm2.72}$ | $\underline{95.6}_{\pm0.02}$ | $35.3_{\pm1.95}$ |
| | ✓ | ✓ | $\mathbf{92.8}_{\pm0.64}$ | $90.2_{\pm0.23}$ | $\mathbf{92.6}_{\pm0.20}$ | $88.3_{\pm1.66}$ |
| AE-KD | ✗ | ✓ | $84.2_{\pm1.52}$ | $61.0_{\pm4.45}$ | $\underline{95.6}_{\pm0.02}$ | $36.9_{\pm0.39}$ |
| | ✓ | ✓ | $91.9_{\pm1.89}$ | $89.0_{\pm3.59}$ | $92.3_{\pm0.04}$ | $89.4_{\pm1.11}$ |
| AGRE-KD | ✗ | ✓ | $84.9_{\pm1.40}$ | $\underline{66.3}_{\pm4.76}$ | $94.5_{\pm3.22}$ | $\underline{39.2}_{\pm0.78}$ |
| | ✓ | ✓ | $91.4_{\pm1.99}$ | $\mathbf{91.1}_{\pm2.56}$ | $91.1_{\pm0.09}$ | $\mathbf{91.9}_{\pm1.17}$ |

Table 3: **Results on self-distillation**. The student and the teacher models have the same network architecture (ResNet-50). The student's worst group test accuracy increases when we distill to a network with higher capacity.

significantly improve the worst-group test accuracy. The gap between the teacher models and students is reduced compared to KD settings in Table 2, where the students' models have smaller capacity. The results indicate that the students' higher capacity can help the network learn more core features and reduce the influence of spurious features in the last layer. On the other hand, AGRE-KD outperforms other baselines, showing that our adaptive weighting scheme effectively guides the student models to focus on minimizing

worst-case group error during training. We further illustrate the effectiveness of our weighting scheme in the next experiment by adjusting the number of debiased teachers in the ensemble.

### 5.2.4 Effect of the number of debiased teachers.

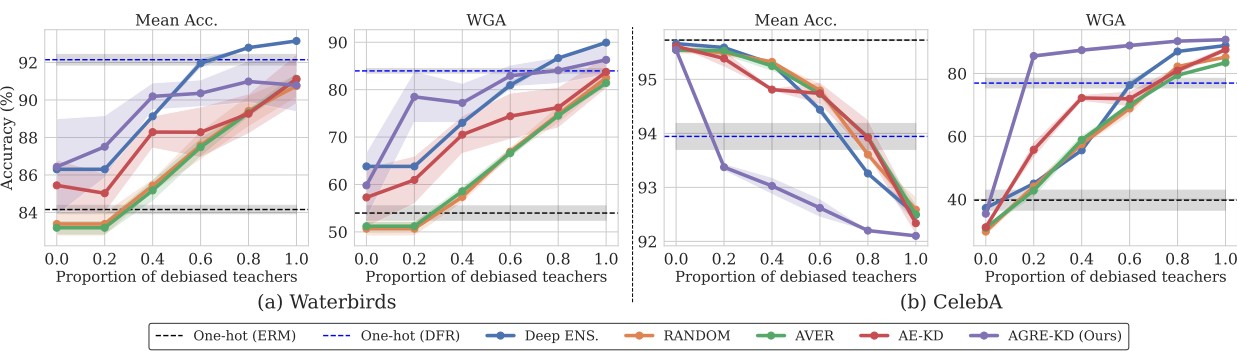

Figure 3: **Results on the proportion of debiased teachers in the ensemble**. We trained the student model using an ensemble of 5 teachers with different ratios of debiased teachers within the ensemble ({0.2, 0.4, 0.6, 0.8, 1}). AGRE-KD effectively upweights and favors the least biased teachers in the ensemble, while other ensemble methods rely more on biased teachers' output and decrease the WGA. AGRE-KD maintains significantly higher WGA, despite having only a single debiased model in the ensemble.

In this experiment, we study how an ensemble with different proportions of biased and debiased models impacts the WGA of the student model. We use the same training process as previously and consider an ensemble of five teachers with different proportions of debiased teachers (i.e., {0.2, 0.4, 0.6, 0.8, 1}); we report the average and WGA in Figure 3 for the Waterbirds and CelebA datasets. The results show that students' WGA increases as the proportion of debiased models in the ensemble increases. When the ensemble contains a single debiased model, the aggregation process of other ensemble KD methods relies more on the majority of biased teachers, leading the student model to capture spurious correlation. On the other hand, our method can implicitly identify and adaptively rely on the knowledge of debiased teachers while reducing reliance on biased teachers. This is reflected by the significantly higher WGA in settings where the ensemble contains a single debiased teacher. Additionally, AGRE-KD outperforms or matches the performance of the deep ensembling model, where the last layer of each model in the ensemble is directly retrained.

### 5.2.5 Ablation on teacher architecture heterogeneity

| Models | Architecture | KD | Debiased Teachers | | ERM Teachers | |
|---|---|---|---|---|---|---|
| | | | Avg | WGA | Avg | WGA |
| Teacher | ViT | ✗ | $96.1_{\pm0.30}$ | $90.1_{\pm1.50}$ | $92.7_{\pm0.00}$ | $73.8_{\pm1.50}$ |
| | ResNet-50 | ✗ | $94.3_{\pm0.8}$ | $89.5_{\pm2.5}$ | $88.7_{\pm0.09}$ | $68.0_{\pm2.20}$ |
| | ConvNeXtV2 | ✗ | $92.4_{\pm0.6}$ | $85.7_{\pm1.5}$ | $86.4_{\pm0.41}$ | $62.0_{\pm0.55}$ |
| RANDOM | | ✓ | $\mathbf{94.3}_{\pm0.18}$ | $80.6_{\pm2.30}$ | $88.9_{\pm0.1}$ | $57.7_{\pm3.21}$ |
| AVER | ResNet-18 | ✓ | $94.2_{\pm0.21}$ | $80.7_{\pm1.10}$ | $84.2_{\pm0.4}$ | $41.7_{\pm2.51}$ |
| AE-KD | | ✓ | $93.5_{\pm0.65}$ | $84.9_{\pm2.81}$ | $89.1_{\pm1.4}$ | $57.9_{\pm1.90}$ |
| AGRE-KD | | ✓ | $93.5_{\pm0.63}$ | $\mathbf{87.9}_{\pm1.25}$ | $\mathbf{89.5}_{\pm0.7}$ | $\mathbf{61.6}_{\pm2.10}$ |

Table 4: Results with heterogeneous teacher ensemble on the Waterbirds dataset. We train teachers with different architectures (ViT (base), ResNet-50, and ConvNeXtV2 (base)) and perform knowledge distillation to a ResNet-18 student model.

Experiments in Section 5.2.3 revealed that the capacity of the student model plays an important role in bias amplification of ensemble KD. Prior work has demonstrated that increasing the diversity of the ensembles results in better generalization performance. We enforced diversity by initializing the weights with independent random seeds (Allen-Zhu & Li, 2020). We now explore the heterogeneity of teacher models by building ensembles with diverse network architectures and analyzing the impact on the WGA accuracy of the student models. We train teacher models using three different network architectures, i.e., ViT (Dosovitskiy et al., 2020), ConvNeXtV2 (Woo et al., 2023), and ResNet-50 (He et al., 2016). Using the same experimental setup as previously, we built an ensemble of nine models with an equal proportion of each model architecture, i.e., three models with each architecture, all trained using different random initialization. This ensures diversity between and within teacher architectures.

The results in Table 4 show, for the Waterbirds dataset, the average and WGA accuracy when the heterogeneous ensemble contains ERM-trained or debiased teacher models. We observe that even under a more heterogeneous teacher ensemble, student models trained with classical ensemble KD methods can match the teacher's average accuracy but incur a significant drop in WGA. Compared to student models trained with homogeneous ensembles of ResNet-50 models (Table 2), we observe that using a more diverse teacher's backbone marginally improves performance across baselines, yet the drop in WGA remains important across ensemble KD methods. This highlights that the bias amplification in ensemble KD does not necessarily depend on ensemble diversity, further supporting our hypothesis of student capacity as a source of the disparity. On the other hand, the orchestration process in AGRE-KD shows better performance as its distillation process encourages better WGA. Despite the teachers' architecture being heterogeneous, we use a ResNet-50 biased model to compute teachers' weights in AGRE-KD during the distillation process, demonstrating the robustness of the proposed approach to the choice of teacher model architecture. The results on the CelebA dataset show the same trends, which we provide in Table 8 in the Appendix.

## 6 Limitation and conclusion

In this paper, we studied bias in ensemble knowledge distillation (KD) and demonstrated that, unlike deep ensemble models that reduce bias, traditional ensemble KD methods can amplify it. We proposed AGRE-KD, an adaptive gradient-based weighting method that improves group robustness in ensemble KD by guiding the student model to learn core features and boosting worst-group accuracy. Our experiments across several benchmarks demonstrated the effectiveness of our approach in distilling knowledge with reduced spurious correlations. While our results highlight AGRE-KD's advantages over existing methods, several limitations and open questions remain. Specifically, the study focused on unsupervised KD using the teachers' logits alone, without access to group or class labels. Although our primary interest was in examining the ability of an ensemble of teachers to distill more robust knowledge, exploring scenarios where class labels are available could enhance the performance of the student further. Moreover, we observed that the WGA improvement in the proposed method is less pronounced when all teachers in the ensemble are biased. This indicates an opportunity to leverage class labels in this context to further enhance group robustness, e.g., by considering the teachers' misclassifications. Finally, additional evaluation using more complex datasets, such as health datasets, is necessary to validate the approach across a wider range of applications.

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

## A AGRE-KD: Algorithm and method overview

In Algorithm 1, we provide a high-level description of the AGRE-KD methodology for training and validation of group-robust distilled models. We also provide an overview of our method in Figure 2.

---

**Algorithm 1** Adaptive Group Robust Ensemble Knowledge Distillation (AGRE-KD)

---

1: **Input:** Ensemble of pretrained teachers $T = \{T_1, T_2, \ldots, T_M\}$, biased model $T_b$, student model $S$ with parameters $\theta$, dataset $\mathcal{D}$, distillation coefficient $\alpha$, temperature parameter $\tau$

2: **for** each minibatch $\{(x_i, y_i)\}_{i=1}^{B}$ from $\mathcal{D}$ **do**

3:      **for** each teacher $T_t \in T$ **do**

4:         Compute knowledge distillation loss $\ell_i^t(\theta)$ for sample $i$ between $S$ and $T_t$       $\triangleright$ Equation 1.

5:         Compute biased model distillation loss $\ell_i^b(\theta)$ for sample $i$ between $S$ and $T_b$       $\triangleright$ Equation 1.

6:         Compute gradient alignment $G_i^t = \langle \nabla \ell_i^t(\theta), \nabla \ell_i^b(\theta) \rangle$ $\triangleright$ Dot product of normalized gradient vectors. Table 6 shows the importance of ignoring the magnitude of the gradients.

7:         Compute the adaptive sample weight for teacher $t$ on sample $i$: $W_t(x_i) = 1 - G_i^t$;

8:      **end for**

9:      Compute weighted knowledge distillation loss:

$$\mathcal{L}_{\text{wKD}} = \frac{\sum_t W_t(x_i) \cdot \ell_i^t(\theta)}{\sum_t W_t(x_i)}$$

10:      Compute classification loss (if labeled data available): $\mathcal{L}_{\text{cls}}$

11:      Compute final loss:

$$\mathcal{L} = \alpha \mathcal{L}_{\text{wKD}} + (1 - \alpha)\mathcal{L}_{\text{cls}}$$

12:      Update student model parameters $\theta$ using gradient descent on $\mathcal{L}$

13: **end for**

---

## B Hyperparameters

We train all models using standard hyperparameters from previous work (LaBonte et al., 2024; Kirichenko et al., 2022; Izmailov et al., 2022) and keep their value fixed across experiments. For the vision tasks, we used an initial learning rate of $1 \times 10^{-3}$ with a cosine learning rate scheduler; we used a batch size of 32 and 100 for the Waterbirds and the CelebA datasets, respectively. For the CivilComments dataset, we use an initial learning rate of $1 \times 10^{-5}$ with a linear learning rate scheduler, a batch size 16, and train for ten epochs. We keep all hyperparameters fixed to train the teacher and student models. For the optimizer, we used AdamW (Loshchilov et al., 2019) and SGD for the language and vision datasets, respectively, with a weight decay of $1 \times 10^{-4}$. Our implementation uses PyTorch (Paszke et al., 2017; 2019), Torch Lightning (Falcon & team, 2019), and Milkshake (LaBonte, 2023).

## C Ablation studies

### C.1 Ablation on $\alpha$

Equations 3 and 7 define the student training objective as a weighted combination of the cross-entropy loss with ground-truth labels and the ensemble knowledge distillation loss. The hyperparameter $\alpha \in [0, 1]$ controls this weighting: smaller values place more emphasis on the ground-truth loss, while larger values prioritize the distillation signal. Since our primary focus is to study how bias propagates through ensemble KD, all main paper experiments fixed $\alpha = 1.0$, thereby relying entirely on distillation.

To investigate the impact of $\alpha$, we performed an ablation with $\alpha \in \{0.1, 0.3, 0.5, 0.7, 0.9, 1.0\}$ on the Waterbirds and CelebA datasets. We used ensembles of five ResNet-50 DFR teachers and a ResNet-18 student, and report results averaged over three random seeds. Figure 4 presents the trends for both average accuracy and worst-group accuracy (WGA).

We observe that WGA increases nearly monotonically with $\alpha$, indicating that when the KD loss dominates, students inherit more of the debiasing effect from the DFR teachers. For example, in Waterbirds with AGRE-KD, WGA rises from 62.5% at $\alpha = 0.1$ to 84.6% at $\alpha = 1.0$. The best-performing baseline (Average KD) improves from 62.3% to 81.3% across the same range, yet still falls short of the One-hot model trained directly with DFR (83.9% WGA).

These results reinforce two key points: (i) ensemble KD benefits from stronger reliance on distillation when teachers are debiased, and (ii) without dedicated mechanisms such as AGRE-KD, ensemble KD can lag behind simpler approaches like direct DFR training. This highlights the importance of explicitly addressing bias amplification in ensemble distillation.

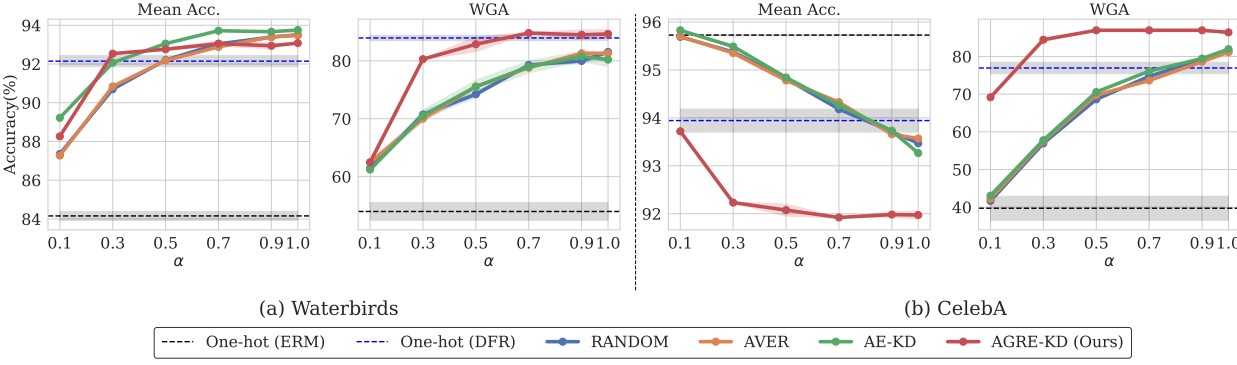

Figure 4: Effect of the parameter $\alpha$ (Eq. 3) on the worst-group accuracy of DFR teachers.

## C.2   Ablation on temperature parameter

Our experiments using the DFR teacher models revealed that reweighting the feature by retraining the last classification layer provides a "dark knowledge" (soft target) to the student model that is more robust to spurious correlation. Since DFR teachers share the same feature representation as their ERM-trained counterparts but differ only in their final classification layer, the enhanced robustness of the student when using DFR teachers indicates that the probability distribution over all classes predicted by the teacher significantly influences the student's dependence on spurious correlations. Since the temperature hyperparameter (Equation 1) softens the probability distribution of the teacher to better expose dark knowledge. We perform an ablation study by varying the temperature parameter ($\tau \in \{1, 2, 4, ..., 10\}$) and analyze its influence on the spurious correlation captured by the student model.

Figure 5 shows the WGA averaged across three independent random seeds in WaterBirds and CelebA datasets. We considered ensembles of five ResNet-50 DFR teachers and a ResNet-18 student. The results show that WGA increases with temperature and then remains steady or decreases after higher values of temperature ($\tau \geq 10$ and $\tau \geq 8$ for the Waterbirds and CelebA datasets, respectively). The smallest WGA is achieved when $\tau = 1$, which means the training uses a regular cross-entropy loss; it increases when we soften the probability distribution ($\tau > 1$) and eventually decreases when the distribution gets almost uniform ($\tau \gg 1$). These findings suggest that the dark knowledge provided by the softened probability distribution of the teacher models reduces students' reliance on spurious correlation during training. These results align with recent work by Mohammadshahi & Ioannou (2024) studying the benefit of increased temperature for fairness in knowledge distillation.

## D   Supplemental results

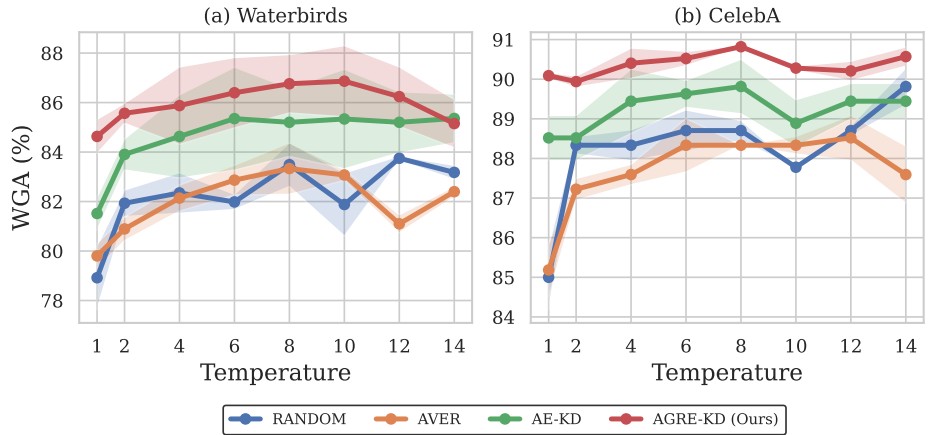

Figure 5: Effect of the temperature parameter on the worst-group accuracy.

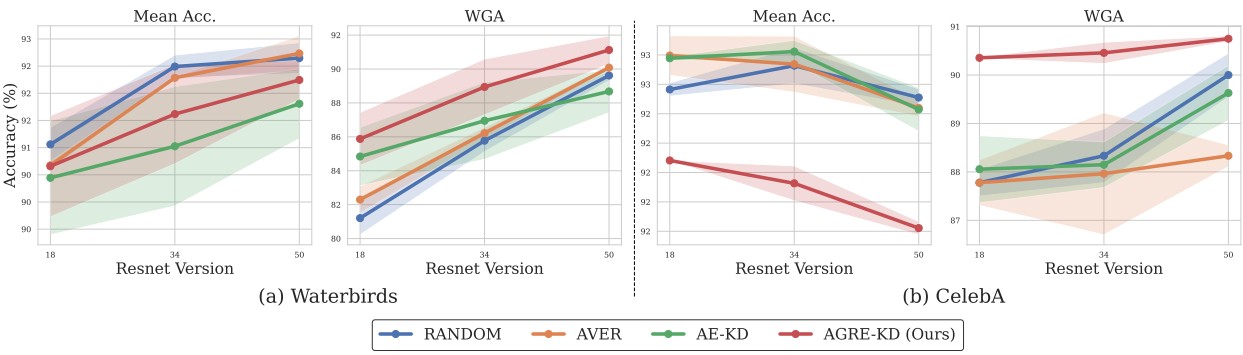

Figure 6: **Results on student model capacity**. We perform experiments on WaterBirds and CelebA using ResNet-50 teachers and varying the capacity of the student network (ResNet-18, ResNet34, and ResNet-50). We plot the average and the worst-case accuracy of different ensemble distillation methods across three random seeds. The WGA tends to increase as the student model has more capacity for learning the core features. Most baselines match or outperform the teachers when the student model has the same capacity as the teachers (i.e., ResNet-50), and our method remains superior in terms of WGA. These results suggest that the reduced capacity of the student model is a source of the disparity observed.

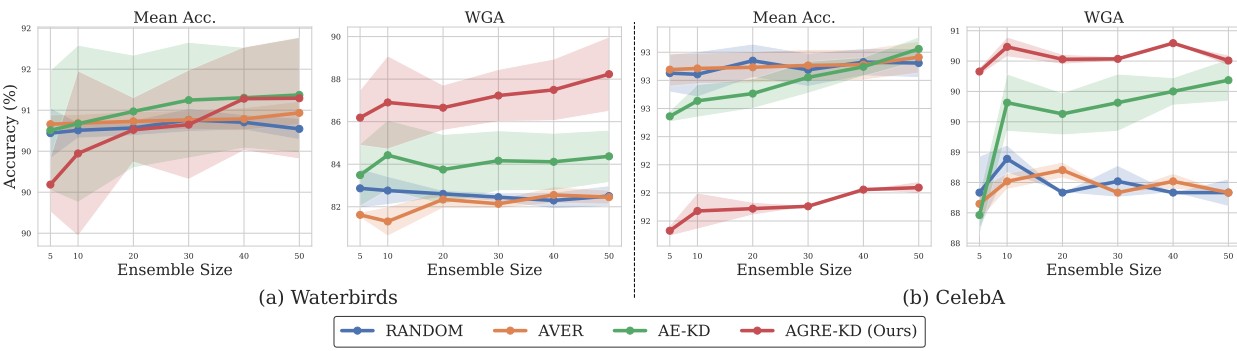

Figure 7: **Effect of ensemble size**. We train the ResNet-18 student model with different ensemble sizes of ResNet-50 teachers (5, 10, 20, . . . , 50). Increasing ensemble size exerts a positive effect on both the average and the worst-group performance. However, the Random KD method tends to get worse as we increase the number of teachers.

| Biased model | Waterbirds | | CelebA | |
|---|---|---|---|---|
| | Average | WGA | Average | WGA |
| ResNet-50 | $90.6_{\pm0.49}$ | $86.7_{\pm2.82}$ | $91.8_{\pm0.20}$ | $90.5_{\pm0.24}$ |
| Resnet34 | $90.0_{\pm1.63}$ | $85.2_{\pm3.48}$ | $91.8_{\pm0.14}$ | $90.5_{\pm0.24}$ |
| ResNet-18 | $90.2_{\pm1.10}$ | $86.8_{\pm2.88}$ | $91.5_{\pm0.09}$ | $89.8_{\pm0.16}$ |

Table 5: **Sensitivity of AGRE-KD to the biased model architecture**. In the main paper, we used a biased model with the same architecture as the teacher models (i.e., ResNet-50). We experiment with different network backbones for the biased models in AGRE-KD. The results below do not show significant differences across biased model choices, demonstrating the robustness of the proposed method to the choice of biased model. These results suggest that the gradient direction of any biased pretrained model can provide sufficient guidance for debiased distillation.

| Model | Waterbirds | | CelebA | |
|---|---|---|---|---|
| | Average | WGA | Average | WGA |
| AGRE-KD w/ grad norm | $90.6_{\pm0.49}$ | $\mathbf{86.8}_{\pm1.86}$ | $91.7_{\pm0.20}$ | $\mathbf{90.9}_{\pm0.71}$ |
| AGRE-KD w/o grad norm | $91.0_{\pm1.32}$ | $81.5_{\pm4.07}$ | $93.1_{\pm0.21}$ | $87.9_{\pm0.52}$ |

Table 6: **Training AGRE-KD using gradient direction with (w/) and without (w/o) gradient normalization**. On the Waterbirds and CelebA datasets, we study how using the gradient magnitude in the weighting scheme impact the results. We trained our AGRE-KD method without normalizing the gradient vectors in the dots product, i.e., accounting for the gradient magnitude of the losses. The results below show that accounting for the gradient magnitude in the weighting scheme reduces the WGA performance. This shows the importance of using normalized dot products to compare directional changes in gradients, making the computed weights independent of the gradient scales themselves, which are generally very noisy.

| Datasets | Sub Groups | #Samples | ERM | Teacher | Random | AVER | AEKD | AGRE-KD |
|---|---|---|---|---|---|---|---|---|
| Waterbirds | (`landbirds`,`land`) | 3498 | $99.3_{\pm0.13}$ | $95.5_{\pm0.97}$ | $96.7_{\pm0.57}$ | $96.8_{\pm0.62}$ | $96.1_{\pm1.29}$ | $94.3_{\pm1.46}$ |
| | (`landbirds`,`water`) | 184 | $74.0_{\pm2.54}$ | $94.3_{\pm0.96}$ | $86.2_{\pm2.20}$ | $87.4_{\pm0.91}$ | $86.4_{\pm3.88}$ | $87.5_{\pm3.14}$ |
| | (`waterbirds`,`land`) | 56 | $54.1_{\pm1.80}$ | $92.1_{\pm1.39}$ | $82.3_{\pm1.65}$ | $82.1_{\pm1.06}$ | $84.6_{\pm3.37}$ | $86.7_{\pm2.82}$ |
| | (`waterbirds`,`water`) | 1057 | $93.4_{\pm0.72}$ | $90.9_{\pm0.81}$ | $91.5_{\pm1.08}$ | $90.9_{\pm0.67}$ | $91.4_{\pm1.97}$ | $90.6_{\pm1.85}$ |
| CelebA | (`nonblond`,`female`) | 71629 | $96.0_{\pm0.39}$ | $91.1_{\pm0.37}$ | $91.8_{\pm0.52}$ | $91.8_{\pm0.52}$ | $91.3_{\pm0.36}$ | $90.4_{\pm0.77}$ |
| | (`nonblond`,`male`) | 66874 | $99.5_{\pm0.06}$ | $92.9_{\pm0.20}$ | $94.3_{\pm0.37}$ | $94.2_{\pm0.20}$ | $93.7_{\pm0.40}$ | $92.4_{\pm0.53}$ |
| | (`blond`,`female`) | 22880 | $85.2_{\pm1.21}$ | $94.4_{\pm0.30}$ | $93.8_{\pm0.35}$ | $93.9_{\pm1.13}$ | $94.6_{\pm0.34}$ | $95.0_{\pm0.83}$ |
| | (`blond`,`male`) | 1387 | $32.3_{\pm2.17}$ | $90.1_{\pm0.86}$ | $88.3_{\pm0.78}$ | $87.5_{\pm0.52}$ | $89.4_{\pm1.63}$ | $91.6_{\pm0.78}$ |

Table 7: **Group-wise accuracy comparison**. We report the group-wise accuracy of different ensemble KD methods on the Waterbirds and CelebA datasets. As in previous experiments, we average the performances across three different random seeds and considered ensembles of five teachers.

| Models | Architecture | KD | Debiased Teacher | | ERM Teacher | |
|---|---|---|---|---|---|---|
| | | | Avg | WGA | Avg | WGA |
| Teacher | ViT | ✗ | $93.9_{\pm 0.48}$ | $85.0_{\pm 1.46}$ | $95.8_{\pm 0.00}$ | $38.8_{\pm 0.78}$ |
| | ResNet-50 | ✗ | $93.2_{\pm 0.92}$ | $90.1_{\pm 0.86}$ | $95.8_{\pm 0.17}$ | $42.2_{\pm 3.14}$ |
| | ConvNeXtV2 | ✗ | $92.9_{\pm 0.07}$ | $86.9_{\pm 0.39}$ | $95.9_{\pm 0.05}$ | $44.4_{\pm 0.78}$ |
| RANDOM | | ✓ | $94.1_{\pm 0.48}$ | $79.4_{\pm 3.14}$ | $95.9_{\pm 0.07}$ | $36.3_{\pm 0.39}$ |
| AVER | ResNet-18 | ✓ | $94.1_{\pm 0.38}$ | $80.0_{\pm 3.10}$ | $95.9_{\pm 0.09}$ | $36.1_{\pm 0.00}$ |
| AE-KD | | ✓ | $93.9_{\pm 0.19}$ | $82.2_{\pm 1.57}$ | $95.9_{\pm 0.10}$ | $40.2_{\pm 1.17}$ |
| AGRE-KD | | ✓ | $90.4_{\pm 0.69}$ | $87.6_{\pm 0.94}$ | $95.9_{\pm 0.03}$ | $41.2_{\pm 1.57}$ |

Table 8: Results with heterogeneous teacher ensemble on the CelebA dataset. We train teachers with different architectures (ViT (base), ResNet-50, and ConvNeXtV2 (base)) and perform knowledge distillation to a ResNet-18 student model.

