# OpenReview forum: "Adaptive Group Robust Ensemble Knowledge Distillation"
_TMLR — Accepted by TMLR_

### Review · Reviewer_6dLt · 2025-08-05

**Summary Of Contributions:**

This paper introduces a novel adaptive group robust ensemble knowledge distillation (AGRE-KD) method designed to address the shortcomings of traditional ensemble knowledge distillation in handling group bias. The core idea is to use a biased model as a reference and dynamically adjust the weights of the teacher models based on the cosine similarity between the gradients of the student model's loss with respect to each teacher and the biased model. This selective distillation helps the student model overcome group bias. The experimental results demonstrate the effectiveness of this method on multiple benchmark datasets. The research has strong practical significance and is highly innovative. The paper is well-written, logically structured, and the experiments are relatively comprehensive. I recommend this paper for publication after some revisions.

## Major Strengths:

- High Innovation: The core idea of the paper is very novel. Using a biased model as "negative guidance" and comparing gradient directions to filter and weight the knowledge from teacher models is an original approach to help students overcome group bias. This method provides a new perspective for the field of knowledge distillation, especially in improving robustness.
- Clear Problem Definition: The paper clearly identifies the limitations of existing ensemble knowledge distillation methods in improving group robustness and empirically validates this issue, laying a solid foundation for the proposed solution.
- Solid Methodology: The theoretical basis and implementation details of the AGRE-KD method are well-explained. In particular, the formula for the gradient-based weighting scheme, is concise and intuitive, perfectly embodying the method's core idea.
- Comprehensive Experiments: The paper conducts thorough experiments on three different datasets (two visual tasks and one language task), using the key metric of worst-group accuracy (WGA) to evaluate model robustness, which makes the conclusions more convincing.
- High-Quality Writing: The paper is well-written and logically structured. Figure 1 (first cited) intuitively illustrates the working principle of the adaptive weighting mechanism through vector directions, while Figure 2 (first cited) clearly outlines the overall flow of the AGRE-KD method. These figures greatly assist readers in understanding the approach.

## Major Weaknesses:
- Deeper Theoretical Analysis Needed: While the paper empirically proves the effectiveness of AGRE-KD, a more in-depth theoretical discussion is needed on why the similarity of gradient directions effectively captures model bias. For example, can it be theoretically proven that this weighting mechanism minimizes the risk on the worst-performing group?
- Sensitivity to the Number of Teachers: The experiments in the paper use a fixed number of teacher models (e.g., 10 for the Waterbirds dataset). Is the performance of the AGRE-KD method sensitive to the number of teachers? How does the method perform with a smaller or larger number of teachers? Exploring this would help to better understand the method's applicability.
- In the Related Work section, under Bias in Knowledge Distillation, there is a punctuation error. The sentence "However, their method also requires class labels to derive the misclassified samples.." has two periods. Please check and correct any similar errors.
- Analysis with Labeled Data:The paper uses unsupervised methods to learn knowledge directly from the teacher model. It is recommended that the author further investigate how the model can utilize label information to mitigate or correct biases in the case of labeled supervision.
- Insufficient Methodological Detail:The methodology section of the paper is brief, making it difficult to clearly understand the model's construction and training process. It is recommended to expand this section and provide more details.

**Audience:**

Yes

**Audience Explanation:**

Interesting paper.

**Claims And Evidence:**

Yes

**Claims Explanation:**

Good paper.

**Requested Changes:**

See the weaknesses part.

---

> ### Author Response · Authors · 2025-09-16
> **Response to Reviewer 6dLt**
>
> We sincerely thank the reviewer for the thoughtful and encouraging feedback. We are pleased that you found the problem definition clear, the methodology novel, and the experiments convincing. Below, we address each of your concerns. All changes have been incorporated in the revised manuscript (highlighted in blue).
>
> **1. Deeper Theoretical Analysis**
>
> > _While the paper empirically proves the effectiveness of AGRE-KD, a more in-depth theoretical discussion is needed on why the similarity of gradient directions effectively captures model bias. For example, can it be theoretically proven that this weighting mechanism minimizes the risk on the worst-performing group?_
>
> We appreciate this suggestion. Our current contribution emphasizes **theoretical intuition supported by strong empirical evidence**. Specifically, the key idea is that gradient alignment with a biased ERM model correlates with overfitting to spurious features. By upweighting teachers whose gradients deviate most from this biased direction, AGRE-KD implicitly guides the student toward representations that better capture core features.
>
> While we stop short of a formal proof, our experiments provide strong empirical support:
>
> - **Figure 3** shows that AGRE-KD consistently improves worst-group accuracy even when only a fraction of the teachers are debiased.
>
> - Across **Tables 1–4**, AGRE-KD achieves the strongest robustness to spurious correlations, outperforming other ensemble KD approaches in both extreme and milder bias regimes.
>
>
> We agree that a rigorous theoretical treatment of gradient similarity and worst-case risk would be valuable, and we identify this as an important avenue for future work.
>
> **2. Sensitivity to the Number of Teachers**
>
> > _The experiments in the paper use a fixed number of teacher models (e.g., 10 for the Waterbirds dataset). Is the performance of the AGRE-KD method sensitive to the number of teachers?_
>
> We have added an ablation in **Appendix Figure 6** analyzing different ensemble sizes. Results show that both average and worst-group performance increase with more teachers across all methods. Importantly, **AGRE-KD maintains stronger worst-group accuracy regardless of ensemble size**, indicating robustness to this factor. This complements our proportion-of-debiased-teachers study (Figure 3), further supporting the method’s stability.
>
> **3. Typos**
>
> > _In the Related Work section, under Bias in Knowledge Distillation, there is a punctuation error. The sentence "… misclassified samples.." has two periods._
>
> We thank the reviewer for catching this. We have corrected the typo and carefully proofread the manuscript to remove similar errors.
>
> **4. Analysis with Labeled Data**
>
> > _It is recommended that the author further investigate how the model can utilize label information to mitigate or correct biases in the case of labeled supervision._
>
> We agree this is an exciting direction. Our focus in this paper is on the **label-free scenario**, which we consider a stricter and underexplored setting. Nevertheless, incorporating label supervision could further enhance robustness. We now discuss this explicitly in the **Limitations and Conclusion (Section 6)**, noting that extending AGRE-KD to supervised contexts (e.g., by leveraging teacher misclassifications or group-labeled subsets) is a promising line of future work.
>
> **5. Methodological Detail**
>
> > _The methodology section of the paper is brief, making it difficult to clearly understand the model's construction and training process. It is recommended to expand this section and provide more details._
>
> We have substantially expanded the methodology description.
>
> - Section 2 (Background) now provides a self-contained overview of knowledge distillation, ensemble KD, and spurious correlations, clarifying how AGRE-KD builds upon these.
>
> - Section 4 (Method) explicitly details the role of the biased model, the computation of gradient-based weights (Eq. 5), and the adaptive aggregation procedure (Eq. 6–7).
>
> - Figures 1 and 2 were revised to more clearly illustrate the gradient-based weighting mechanism and training workflow.
>
>
> ---
> We thank the reviewer again for their careful review and detailed feedback, which have helped improve the paper's clarity. We hope the revised paper fully addresses the concerns raised and we remain available to address any remaining concerns.

---

### Review · Reviewer_2r2L · 2025-08-06

**Summary Of Contributions:**

The paper studies the situation where knowledge distillation (i.e. a smaller-capacity neural net learning to reproduce a large capacity network) is applied to an ensemble of teachers on a domain where there are spurious correlations between features and the outputs (e.g. the background natural scene in a waterbirds vs. landbirds discrimination task). Knowledge distillation on ensembles can worsen error rates on rare subgroups (e.g. pictures of waterbirds on land, where they normally do not live). The authors present a method to prevent this poor performance on rare subgroups. They use a biased model (one which has learned the spurious correlation) to guide the student network away from the biased model, by guiding it towards the gradient of the teacher in the ensemble which is most different from that of the biased model. The authors also show that knowledge distillation is the key driver of the poor performance on rare subgroups, by showing that the problem does not occur when the student has the same capacity as the teacher. The authors also discuss a method for retraining the last layer only of teachers so as to mitigate bias. Positive results (as compared to competing methods) on 3 benchmarks are presented.

A key strength of the paper is the clear win by their method over competing methods on the 3 benchmarks. The method is also novel, to the best of my knowledge.

One significant weakness is the failure of the paper to situate the research in the context of preexisting covariate shift research, such as the work of Sugiyama. Relatedly, the authors refer many times to spurious correlations without defining 'spurious' in a rigorous way.

Another concern is that while the results in some ways are strong, they are not as broad-based as I would like. 3 benchmarks seems like to small-to-moderate amount of evidence for a journal paper, where there is more time for authors to generate evidence than for a conference paper.

**Additional Comments:**

Typos:

Page 1: but not causally related -> but are not causally related
Page 2: whose only the last layer -> whose last layer was the only one
Page 3: using a given the training -> using a training set
Page 3: of the a given sample -> of the given sample
Page 6 improve the performances-> improve the performance
Page 6 word group accuracy -> worst group accuracy
Page 7 for the forvision tasks -> for the vision tasks
Page 7 is spuriously correlated gender-> is spurious correlated with gender
Page 9 can consistently improves -> improve

**Audience:**

Yes

**Audience Explanation:**

Yes, while the topic is fairly specific (knowledge distillation from an ensemble), some readers would be interested.

**Broader Impact Concerns:**

No concerns. This is a good paper from the point of view of trying to prevent harm to rare and overlooked subgroups.

**Claims And Evidence:**

Yes

**Claims Explanation:**

I think I may be able to get to the point where I would answer 'yes' after a revision. However, for now, I would answer 'no', partly because I think 3 benchmarks is not quite enough.

I also don't feel comfortable accepting this paper as-is without an explanation from the authors about how their research relates to the evidence on covariate shift.

To illustrate the importance of thinking in terms of covariate shift, let's look at the waterbirds vs. landbirds example. I concede that ideally, a perfect classifier of waterbirds vs. landbirds would only look at the traits of the birds themselves and not the backgrounds of the pictures. However, suppose perfect classification is not possible based on the apperance of the birds themselves...maybe the birds are partially occluded in some pictures, or maybe the pictures are low-resolution enough that you can't always distinguish based on the appearance of the birds themselves. Suppose also that we can be confident that we will only need to classify photos taken from nature rather than e.g. from a zoo where a waterbird might appear against a land background. In this scenario, the optimal classifier probably would use the background...the correlation, while technically spurious, would nonetheless improve classification performance, assuming that the input distribution did not change, i.e., that we continued to classify natural images. It would only be under covariate shift, where waterbirds appearing on land actually starts occurring at a significant non-zero rate, that we would want to avoid using the 'spurious' correlation.

**Requested Changes:**

1) Positive results on at least one more benchmark.

2) Discussion of the covariate shift literature and how it relates to this research.

3) A little more rigor and detail on what it means for a student to have smaller 'capacity' than a teacher. I imagine it's usually assumed that the student has both fewer hidden layers and fewer hidden units per layer. However, the paper should try to define it more carefully. Does a student network with 5 hidden layers but 1000 hidden units per layer have a smaller or larger capacity than a teacher network with 10 hidden layers but only 50 hidden units per layer? Are machine learning theory concepts like VC dimension relevant here?

---

> ### Author Response · Authors · 2025-09-16
> **Response to Reviewer 2r2L**
>
> We thank the reviewer for the constructive and detailed feedback. We are pleased that you found our method novel and our results compelling, and we have revised the manuscript to address your main concerns. All changes are highlighted in blue.
>
> **1. Additional Benchmark**
>
> > _Positive results on at least one more benchmark._
>
> We have added experiments on **Colored MNIST**, a synthetic dataset widely used to study spurious correlations. This dataset extends MNIST by associating each digit with a color distribution, creating bias-aligned and bias-conflicting samples. We vary the strength of this correlation ({99.5%, 99%, 98%, 95%}) to test robustness under both extreme and mild bias regimes.
>
> The new results (Section 5.2.1, **Table 1**) show that:
>
> - Debiased teachers are essential for transferring robust knowledge.
>
> - While deep ensembles improve WGA, classical ensemble KD can degrade it under strong bias.
>
> - AGRE-KD consistently outperforms all baselines, maintaining high average accuracy while significantly improving worst-group accuracy across regimes.
>
>
> This additional benchmark strengthens our empirical evidence and confirms the generality of our findings.
>
>  **2. Spurious Correlations and Covariate Shift**
>
> > _Discussion of the covariate shift literature and how it relates to this research._
>
> We have expanded **Section 2** to provide a more rigorous definition of spurious correlations: _non-causal features (e.g., background) that correlate but is not predictive of the target feature_. Models often exploit such shortcuts, which harms minority subgroups and fairness [1].
>
> We also added a **discussion on covariate shift** (Section 2). Covariate shift arises when the marginal distribution of inputs changes between training and test, while the conditional label distribution remains fixed. In contrast, spurious correlation describes a training scenario where the model relies on features that are **never causally predictive**, even without distribution shift. The two notions could intersect when the “spurious” feature itself is the source of the shift. We now clarify this distinction and cite covariate shift literature (e.g., Sugiyama’s work), better positioning the paper: instead of correcting distribution mismatches, our method discourages reliance on spurious features during training.
>
> **3. Model Capacity**
>
> > _A little more rigor and detail on what it means for a student to have smaller ‘capacity’ than a teacher._
>
> We have clarified this in **Section 5.2.3**. By “capacity,” we mean the **number of trainable parameters and representational richness**. For example, our teacher models (ResNet-50, ~25.6M parameters) are substantially larger than student models (ResNet-18, ~11M). This difference explains why self-distillation (same-capacity teacher and student) does not suffer the same degradation: larger-capacity models can represent finer-grained features, whereas smaller students amplify spurious shortcuts when distilling from ensembles. We also note in the revision that formal notions of capacity (e.g., VC dimension) could be relevant, but we leave a rigorous theoretical aspect to future work.
>
>
>  **4. Typos**
>
> > _Corrections to phrasing and minor errors (e.g., “but not causally related → but are not causally related”)._
>
> We carefully proofread the manuscript and corrected all listed typos.
> These revisions strengthen the paper’s positioning and address the reviewer’s concerns about scope, rigor, and clarity.
>
> ---
>
> We thank the reviewer again for their careful review and detailed feedback, which has helped improve the paper's clarity. We hope the added results on an additional benchmark and the revised paper fully address the concerns raised, and we remain available to address any remaining concerns.
>
> **Reference**
>
> [1] Ye, Wenqian, et al. "Spurious correlations in machine learning: A survey." _arXiv preprint arXiv:2402.12715_ (2024).

---

> > ### Comment · Reviewer_2r2L · 2025-10-09
> > **Thanks for the experiment and revisions**
> >
> > In light of the additional experiment and additional discussion of covariate shift and model capacity, I will change my 'claims' answer to 'yes' and recommend acceptance.

---

### Review · Reviewer_tNEq · 2025-09-03

**Summary Of Contributions:**

This paper proposes a study on the biases arising in ensemble knowledge distillation as well as a method to mitigate them, based on following the gradient direction in the opposite direction of a biased teacher. Experiments are conducted on simple vision and text datasets such as Waterbirds, CelebA and CivilComment, using simple architectures such as Resnets and BERTs, demonstrating the biases and the effectiveness of the approach.

**Audience:**

Yes

**Audience Explanation:**

Distilling knowledge from multiple teachers is a practical and valuable approach in deep learning. Although bias has rarely been studied in this context—except in deep ensembles—it turns out that bias can still exist even when all teachers are debiased. This finding alone is quite interesting. Moreover, the paper goes further by proposing a solution to address this issue. This work will be of interest to practitioners.

**Claims And Evidence:**

Yes

**Claims Explanation:**

Strenghts:

- The experiments show in a vision and a language setup, that distilling from multiple debiased teachers can lead to a biased student. The experiments also demonstrate that the proposed teacher weighting mechanism is effective and produces a less biased student.

- Additional ablations on model capacity and architecture and numbers of teachers are conducted and further validate the conclusions.

Weaknesses:

- Overall I found the description of the different experimental setups unclear. The information is often either scattered across different sections in the text, or incomplete:

- The presentation of Table 1 and of the baselines could be improved. It would be more clear if one bullet point in the text corresponds to one row in the table, in the same order. Right now “one-hot” is barely mentioned in a very unclear way, “AVER” is before “random” in text but appears after in the text, and the lines separating the rows in the table are inconsistent.

- It is not very clear for each experiment how the biased model is obtained. Section 4 mentions “AGRE-KD relies on a model pretrained with ERM that captured the dataset’s spurious correlation (biased model)” but more details can only be found later in the experimental section. I believe those details are very important and should already be described in Section 4.

- Only later the paper mentions: “For the biased model used by our method to compute teachers’s weights, we randomly select one teacher model trained with ERM and without DFR” But does this mean that some teachers are trained with DFR and some others are not ? Can you provide more details on that ?

- The experimental setup is a bit outdated/toyish. The vision models used are old ResNet-18 and ResNet-50. Table 3 presents results with ViT and ConvNeXt v2 but does not mention the model size. The text models are also old: BERT and DistillBERT are not much used nowadays in the era of LLMs. The datasets are also fairly simple and old. Have you thought about a more concrete setup where Ensemble-KD really unlocks a result that was not possible without ?

- The paper mentions: “we consider ERM-trained (biased) and debiased teachers obtained by deep feature reweighting (DFR).” Do you measure how effective this is in practice ?

- In general in the experiments, how is the WGA determined ?

- If my understanding of the “one-hot” baseline is correct, KD, in particular Ensemble-KD should give better results than “one-hot”, otherwise what is the point of distillation ? The experiments show that this is not the case, do you have an intuition on why ? Maybe the setup is not that interesting ?

- The abstract and introduction of the paper do not mention the experimental protocol or the quantitative gains on the protocol. Good ideas can only be validated by strong experiments and I believe the experiments should be part of the main findings and discussed early in the paper.

- In Section 5.2, when describing the results in the 3 bullet points : it would be much clearer to refer to concrete numbers in the table when describing the findings, rather than only saying “From the results in Table 1” and letting the reader infer the conclusion from the number themselves.

- Small typo in the introduction, there is a missing space: “model.Figure 1 illustrates the gradient”.

**Requested Changes:**

Improve the clarity when describing the method and experimental setups. See weaknesses.

---

> ### Author Response · Authors · 2025-09-16
> **Response to Reviewer tNEq**
>
> We thank the reviewer for the thoughtful feedback and constructive suggestions. We are glad that you found our study of bias in ensemble knowledge distillation both interesting and well supported by experiments, and the proposed weighting mechanism effective. Below, we address each of your concerns point by point. Main revisions are highlighted in magenta in the manuscript.
>
> **1. Clarity of Baseline Descriptions and Table 1**
>
> > _The presentation of Table 1 and of the baselines could be improved. It would be more clear if one bullet point in the text corresponds to one row in the table, in the same order. Right now “one-hot” [...]_
>
> We revised the experimental setup **(Section 5.1)** to describe each baseline in the exact order of Table 1. Each bullet point now maps directly to one table row, improving readability. We also reorganized the table formatting so that lines consistently group methods by whether they use KD.
>
> **2. Biased Model Clarification**
>
> > _It is not very clear for each experiment how the biased model is obtained… does this mean that some teachers are trained with DFR and some others are not?_
>
> We clarified in **Section 4** that the biased model is always **one ERM-trained teacher** randomly selected from the ensemble across runs and seeds. In contrast, **debiased teachers** are obtained via last-layer retraining with DFR. Thus, only in the mixed setting **(Section 5.2.4)**, both ERM-trained and debiased teachers are included in the ensemble, and AGRE-KD adaptively upweights those least aligned with the biased model.
>
> **3. Definition of Worst-Group Accuracy (WGA)**
>
> > _In general in the experiments, how is the WGA determined?_
>
> We now formally define WGA in **Section 2**: it is the test accuracy of the group experiencing the lowest performance, computed after evaluating group-wise accuracy on the test set. This metric quantifies subgroup robustness under spurious correlations.
>
>  **4. One-Hot Baseline vs. Ensemble KD**
>
> > _If my understanding of the “one-hot” baseline is correct, KD, in particular Ensemble-KD should give better results than “one-hot”, otherwise what is the point of distillation?_
>
> We agree this is a key observation. Our results reveal that **classical ensemble KD can amplify bias so strongly that it underperforms the one-hot baseline**, especially under extreme spurious correlations. For example, on **Waterbirds**, Random KD achieves only 39.6% WGA compared to 54.1% for one-hot training.
>
> To further probe this, we added **Colored MNIST experiments (Section 5.2.1, Table 1)**. They show that in extreme bias regimes, one-hot can indeed outperform ensemble KD. However, as bias weakens (e.g., at 95% bias-aligned ratio), ensemble KD improves upon one-hot by up to **+40% WGA**. This highlights that while naive ensemble KD can fail, AGRE-KD consistently avoids this collapse and surpasses both one-hot and competing KD methods.
>
> **5. Abstract, Introduction, and Result Presentation**
>
> > _...Section 5.2 should cite concrete numbers when describing results._
>
> We revised both the **abstract and introduction** to emphasize experimental validation. For example, we now state upfront that AGRE-KD achieves **+3.0% WGA on CelebA** and even outperforms the deep ensemble itself.
>
> We also rewrote **Section 5.2** to cite explicit numbers when describing findings. For instance, instead of “From Table 1 we see…”, we now directly state: “AGRE-KD improves WGA from 82.9% (AVER KD) to 87.9% on Waterbirds.”
>
> **6. Model and Dataset Choices**
>
> > _The experimental setup is a bit outdated/toyish… have you thought about a more concrete setup?_
>
> Our choice of **ResNet-18/50 and BERT/DistilBERT** was guided by prior work on fairness and spurious correlations, to ensure comparability with established baselines. To mitigate this concern, we included **additional experiments with heterogeneous modern architectures (ViT and ConvNeXtV2) in Table 4**, showing that AGRE-KD remains robust even with different backbones. While larger-scale LLMs are beyond our current scope, we will highlight this as an important direction for future work, where bias-aware ensemble KD could be critical.
>
> **7. Effectiveness of DFR Teachers**
>
> > _The paper mentions… debiased teachers obtained by DFR. Do you measure how effective this is in practice?_
>
> Yes. Across all datasets, ensemble KD with debiased teachers yields **substantial WGA improvements** over ERM teachers. For example, on CelebA, WGA rises from 37.5% (ERM teacher) to 90.1% (DFR teacher). We now emphasize these results in Section 5.2, clarifying that DFR plays a key role in ensuring robust knowledge transfer with a complementary ablation study of $\alpha$ in Appendix C.1.
>
> **8. Minor Issues**
>
> We fixed the missing space typo (“model.Figure” -> “model. Figure” and carefully proofread the manuscript for similar issues.
>
> ---
>
> We thank the reviewer again for their thoughtful comments, which significantly improved the clarity and presentation of our work.

---

### Review · Reviewer_SyT8 · 2025-09-04

**Summary Of Contributions:**

This work highlights how standard ensemble distillation methods of teaching a single student model with larger teacher models may lead the student more biased against underrepresented groups with spurious correlations. To address this, the authors propose AGRE-KD that modifies the teacher ensemble's signal to the student by down-weighting the contribution of teachers that align more with a separate biased model. Experimental results across a variety of architectures and classification domains indicate that this method can improve performance upon other ensemble KD methods, especially for the worst-performing groups in the dataset domain.

**Audience:**

Yes

**Audience Explanation:**

In it's current state, this paper is mostly interesting to other members of the ensemble KD and debiasing communities. However, it requires further transparency (both in table highlighting and in qualitative discussion) on the performance compared to standard training without ensemble KD, as discussed both above and below. Why should a practitioner consider debiased ensemble KD if they may be able to get comparable performance much more cheaply just training the target model directly on their training data (possibly with DFR debiasing)?

**Broader Impact Concerns:**

No significant ethical concerns beyond those for standard deep learning research.

**Claims And Evidence:**

Yes

**Claims Explanation:**

Most claims are supported with accurate, convincing, and clear evidence. However, there are some gaps in communicating the full story to the audience.

- It is misleading to not include "One-hot" in your table bolding/underlining scheme. Especially for audience members from outside the distillation community, it is important to see when ensemble KD methods can outperform standard ground truth training. In particular, in Table 1, One-hot's results are consistently beating or within the confidence bounds of the best ensemble KD results.
- The datasets and architectures used came directly from related works, but some are becoming somewhat dated. Also, only repeating previously used datasets and architectures may lead to overfitting on these domains, so we don't know how your findings may generalize.

**Requested Changes:**

Please include "One-hot" in your table bolding/underlining scheme, as detailed above.

Please add an experimental analysis of $\alpha$, since it is introduced in the methods but then set to $1$ without further experimentation (except for "One-hot" technically using $\alpha=0$. As your analysis of $\tau$ shows uniform effects on different algorithms, a similar experimental exploration of $\alpha$ may reveal important results for applying these methods in "the real world".

The different datasets seem to have different effects on the algorithms' results: in CelebA, debiasing the student always results in reduced overall performance and increased WGA performance, while for CivilComment this is much less consistent. Can you provide qualitative discussion of these? Further, are there other datasets (even not used by prior works) that can be studied to provide further understanding of these methods?

Tables 1-2: the column names "Debiased" and "KD" aren't very clear even with the given definitions. When is the student/target model debiased, and when are the teachers debiased?

Figure 3 would be more interesting if you included a ratio of $\frac05$ to see how each algorithm performs with non-debiased models, as well as add a horizontal line for the performance of One-hot in each subplot.

Figure 6: "Effect of teacher size" implies the model size of each teacher, so "Effect of ensemble size" would be more clear.

Typos:
- Period missing in last sentence of Paragraph 2 of Section 2. Background.
- Occasional misuses of `\citet` that should be `\citep`, such as "...by mimicking the teacher model $f^t$ ~Hinton et al. (2015)~(Hinton et al., 2015)."
- Naming consistency: replace all of "resnet50", "Resnet-50", "Resnet50", etc. with one consistent name ("ResNet-50" is used in the original work) both in the text and in plot labels, and check other architecture/dataset/etc. names for similar issues.

---

> ### Author Response · Authors · 2025-09-16
> **Response to Reviewer SyT8**
>
> We sincerely thank the reviewer for the constructive and detailed feedback. We have revised the paper accordingly and summarize the key changes and clarifications below. We highlighted key changes in red in the revised manuscript.
>
> ---
>
> > *Please include "One-hot" in your table bolding/underlining scheme. [...] Why should a practitioner consider debiased ensemble KD if they may be able to get comparable performance much more cheaply just training the target model directly on their training data (possibly with DFR debiasing)?*
>
> We agree this is an important perspective. Our study primarily aims to analyze how ensemble KD methods behave under spurious correlations. For this reason, we highlighted (bolded/underlined) comparisons between ensemble KD methods. The "One-hot" baseline was included as a reference point, but we now explicitly highlight it in the tables for transparency.
>
> We also added a discussion comparing ensemble KD to direct one-hot training. As shown in ablation studies, ensemble KD can surpass one-hot training in certain conditions, e.g., more diverse ensembles such as ensembles with heterogeneous architecture (in Table 4) and lower-bias regimes in the newly added Colored-MNIST). When comparing to the one-hot baseline, we found mixed results, but generally observed that ensemble KD can increase average accuracy compared to the "One-hot" baseline, but they are also more prone to bias amplification.  This reinforces our finding that while ensemble KD offers advantages, debiasing mechanisms are essential under spurious correlation to make it competitive with simpler approaches like direct training or DFR.
>
> > *Please add an experimental analysis of $\alpha$*
>
> Similar to the experimental exploration of the temperature parameter $\tau$, we performed experiments with different values of $\alpha$ (Appendix C.1) showing that across ensemble KD baselines, the worst-group accuracy increases as more weight is given to ensemble knowledge distillation with five DFR teachers, i.e., the student model gets less biased as $\alpha$ increases. For example, on the Waterbirds dataset, the WGA increases linearly (by up to +25%) with increased values of $\alpha$ across ensemble KD methods.
>
>
> > *The different datasets seem to have different effects on the algorithms' results (CelebA vs. CivilComment). Can you provide qualitative discussion? Further, are there other datasets that can be studied to provide further understanding of these methods?*
>
> We expanded the discussion to analyze dataset-specific behavior. For CelebA, skewness in subgroup distribution (e.g., _blond-male_ vs. _nonblond-female_) drives a trade-off between worst-group accuracy and average accuracy, explaining the observed performance drop.
>
> To broaden the scope, we added experiments on **Colored-MNIST** with varying levels of spurious correlation (99.5%, 99.0%, 98.0%, 95.0%). This dataset provides a controlled setting to study how ensemble KD methods respond to different bias regimes. Results show that while ensemble KD is comparable to one-hot and deep ensembles in highly biased settings, it achieves significant gains as bias decreases—underscoring the sensitivity of ensemble KD to dataset characteristics and the role of our proposed method in addressing this.
>
>
> > *Tables 1–2: the column names "Debiased" and "KD" aren't very clear. When is the student/target model debiased, and when are the teachers debiased?*
>
> We clarified this in the revision:
>
> - The **Debiased** column indicates whether the **teacher ensemble** consists of debiased models.
>
> - The **KD** column indicates whether the **student** is trained with knowledge distillation (vs. direct one-hot).
>
>
> This distinction is now clearly stated in the table captions and text.
>
>
> > *Figure 3 would be more interesting if you included a ratio [...] and a horizontal line for One-hot.*
>
> We added the requested ratio (0/5) to the plots, which represents performance with non-debiased teachers. We also added a horizontal reference line showing One-hot performance for better interpretability.
>
>
> > *Figure 6: "Effect of teacher size" implies model size, so "Effect of ensemble size" would be clearer.*
>
> We have corrected the title to "Effect of Ensemble Size."
>
>
> > *Typos and formatting issues (e.g., missing period, \citet vs. \citep, naming consistency such as ResNet-50).*
>
> We carefully proofread and fixed all identified issues, ensuring consistency in citations and naming conventions (e.g., always "ResNet-50").
>
> ---
> We thank the reviewer again for their careful review and detailed feedback, which has helped improve the paper's clarity. We hope the revised paper fully addresses the concerns raised, and we remain available to address any remaining concerns.

---

> > ### Comment · Reviewer_SyT8 · 2025-09-17
> > **Response to authors**
> >
> > Thank you for your revisions and response. I see most discussed revisions were made, but you claimed "we now explicitly highlight [the "One-hot" baseline] in the tables for transparency" yet did not do so in what are now Tables 2 and 3.

---

> > > ### Author Response · Authors · 2025-09-18
> > >
> > > Thank you for pointing this out and for carefully rechecking our revision. We have corrected this in the final version: the "One-hot" baseline is now clearly marked in both Tables 2 and 3 to highlight comparison to ensemble KD baselines. We also ensured consistent formatting across all tables to avoid similar confusion.
> > >
> > > We appreciate your attention to detail, which has helped us improve the clarity and consistency of our manuscript.

---

### Decision · Action_Editor_cBvM · 2025-10-16

**Recommendation:** Accept as is

**Additional Comments:**

Minor issues:
- Typo above Eq. 5: should be penalize examples with gradient dot product close to 1 and upweight close to -1?
- Top of page 7, should be \citep for (Kirichenko; Ye), so that the citations are not a part of the sentence.
- Should the KD column for one-hot in Table say “no”? It does say no in tables 2,3, but not 1.

**Audience:**

Yes

**Audience Explanation:**

The paper may be of interest to both people working on knowledge distillation and people working on group robustness.

**Claims And Evidence:**

Yes

**Claims Explanation:**

The paper studies ensemble distillation in the context of group robustness. The authors show that in certain situations ensemble distillation can lead to stronger bias than simple ERM, i.e. distillation can amplify bias. They also propose a method for adaptively weighting the teachers in the ensemble based on their similarity to a separately trained biased model (computed via dot product of the gradients of the loss). They show that knowledge distillation leads to more robust models when the teachers are weighted this way. The experiments are on standard spurious correlation benchmarks,

The claims are well supported by experiments. The main downside highlighted by multiple reviewers is that the experiments are somewhat outdated by now, as they are reusing the same data and architectures as 2020 papers. The authors introduced some experiments with more modern models.